# Myc controls a distinct transcriptional program in fetal thymic epithelial cells that determines thymus growth

Jennifer E. Cowan[1], Justin Malin[1], Yongge Zhao[1], Mina O. Seedhom[2], Christelle Harly[1], Izumi Ohigashi[3], Michael Kelly[4], Yousuke Takahama[5], Jonathan W. Yewdell [2], Maggie Cam[6] & Avinash Bhandoola [1]*

Interactions between thymic epithelial cells (TEC) and developing thymocytes are essential for T cell development, but molecular insights on TEC and thymus homeostasis are still lacking. Here we identify distinct transcriptional programs of TEC that account for their age-specific properties, including proliferation rates, engraftability and function. Further analyses identify Myc as a regulator of fetal thymus development to support the rapid increase of thymus size during fetal life. Enforced Myc expression in TEC induces the prolonged maintenance of a fetal-specific transcriptional program, which in turn extends the growth phase of the thymus and enhances thymic output; meanwhile, inducible expression of Myc in adult TEC similarly promotes thymic growth. Mechanistically, this Myc function is associated with enhanced ribosomal biogenesis in TEC. Our study thus identifies age-specific transcriptional programs in TEC, and establishes that Myc controls thymus size.

[1] Laboratory of Genome Integrity, Center for Cancer Research, National Cancer Institute, National Institutes of Health, Bethesda, MD 20892, USA. [2] Laboratory of Viral Diseases, National Institute of Allergy and Infectious Diseases, National Institutes of Health, Bethesda, MD 20892, USA. [3] Division of Experimental Immunology, Institute of Advanced Medical Sciences, University of Tokushima, Tokushima 770-8503, Japan. [4] Single Cell Analysis Facility, Center for Cancer Research, National Cancer Institute, National Institutes of Health, Bethesda, MD 20892, USA. [5] Experimental Immunology Branch, National Cancer Institute, National Institutes of Health, Bethesda, MD 20892, USA. [6] Office of Science and Technology Resources, Office of the Director, Center for Cancer Research, National Cancer Institute, National Institutes of Health, Bethesda, MD 20892, USA. *email: avinash.bhandoola@nih.gov

During mouse embryogenesis, the thymus undergoes rapid growth, doubling in size daily until birth. After birth, the thymus continues to expand in size, although at a reduced rate, until peaking in size at ~4 weeks of age. This maximal thymus size is maintained in adult mice until puberty, around 8 weeks, after which it begins to decline[1]. As mice age, there is a dramatic reduction in thymic cellularity, decline in thymic output and diminished T cell function[1]. These dynamic changes in thymus size through life are regulated by TEC[2–4] but their molecular basis is poorly understood.

The transcription factor Myc has been identified as controlling organ size in *Drosophila*[5]. In vertebrates, however, a role for Myc in controlling organ growth and size is less clear. In mice, incremental reductions in Myc expression using an allelic series of mouse models resulted in an incremental decrease in total body mass, which was accompanied by hypoplasia of some tissues, including the thymus[6]. Mice with targeted Myc depletion in TEC present with small thymi in adulthood, accompanied by reduced TEC proliferation rates and decreased TEC numbers[7]. Hence Myc is needed in TEC for normal thymus size. However, gain of function experiments with Myc have not been performed, and so it is unknown if Myc activity limits thymic size and function. In addition, it is unknown whether alterations in Myc activity occur with age, and so might contribute to the age-dependent pattern of vertebrate thymus growth.

Here, we use next generation sequencing techniques to contrast transcriptional dynamics between fetal and adult TEC. We determine genes underlying the transcriptional programs that differ between fetal and adult TEC, and identify Myc and downstream Myc target genes as controllers of the rapid expansion in thymus size from embryonic stages until young adulthood. We show that ectopic Myc restores the expression of a Myc-controlled gene module to adult TEC, together with functional properties that are normally specific to embryonic TEC. Our results provide insight into mechanisms that underlie functional differences between fetal and adult TEC. They establish that Myc activity in TEC restricts thymus size through life and provide a clear demonstration that Myc activity controls organ size in vertebrates.

## Results

**Transcriptional diversity of TEC through development**. To identify age-specific transcriptional profiles of TEC, we performed RNA-seq on isolated TEC populations from selected developmental timepoints. TEC comprise cortical TEC (cTEC), and medullary TEC (mTEC)[8]. At E13.5, cTEC, and mTEC subsets cannot be distinguished based on the markers Ly51 and UEA, therefore we isolated total TEC. We isolated cTEC and mTEC from E15.5, E17.5, newborn, adult and aged mice (Gating strategies in Supplementary Fig. 1a and b). Experimental replicates clustered closely, as expected (Supplementary Fig. 1c).

Principal-component analysis (PCA) revealed that PC1 (20% of all variance) separated samples based on lineage (Fig. 1a). Thus, cTEC and mTEC maintain distinct transcriptional identities, regardless of developmental timepoint. PC2 (10% of all variance) separated samples by developmental timepoint. This second component revealed greater transcriptional changes at 2-day increments of cTEC development during embryogenesis to adulthood, compared with changes seen between the 17-month[+] age increase between adult and aged cTEC. We found several thousand genes had significantly altered expression between E13.5 TEC and adult cTEC or mTEC (Supplementary Fig. 1d). Moreover, over a thousand genes upregulated or downregulated with age were shared between the two lineages. In contrast, many fewer genes

demonstrated significantly altered expression between adult and aged cTEC and mTEC (Supplementary Fig. 1e), of which most were lineage-specific, with less than a hundred upregulated and downregulated genes overlapping between cTEC and mTEC.

We determined biological processes associated with the age-dependent diversification of TEC, using gene set variation analysis (GSVA)[9] performed separately on cTEC and mTEC compared against the same E13.5 TEC subset (Supplementary Fig. 1f). The 2000 most differentially expressed genes across all timepoints (Supplementary Data 1) were selected for this analysis[10]. Significantly altered pathways are listed in Supplementary Data 2. Genes highly expressed in both fetal cTEC and mTEC were enriched for pathways involved in cell cycle, ribosomal biogenesis and other described Myc target genes (Fig. 1b, c). Genes continuing to decline in expression between adult and aged cTEC remained enriched for many of these pathways (Supplementary Fig. 1g and Supplementary Data 3). In contrast, pathways enriched in adulthood or upregulated with age included lysosomes and antigen presentation in both cTEC and mTEC (Fig. 1b, c). These data considerably extend previous work demonstrating that cell cycle signatures are enriched in fetal TEC, and antigen processing and presentation pathways are highly enriched in adult TEC[11,12].

We focused on genes that govern pathways highly enriched at the earliest stages in TEC development, which included cell cycle, ribosomal biogenesis, and other genes downstream of Myc, with the presumption that these control the rapid expansion in thymic size observed in early life.

**Increased proliferation and ribosomal biogenesis in fetal TEC**. To examine how transcriptional alterations to cell cycle genes in TEC during thymic maturation correlated with decreased rates of cell division, we performed in vivo single pulse bromodeoxyuridine (BrdU) labeling and measured the frequency of BrdU incorporating TEC 18 h[+] after labeling[13]. At E15.5, ~30% of TEC were BrdU[+], this declined to under 20% in TEC at birth, and decreased to <5% by 6 weeks of age (Fig. 2a).

GSVA identified overrepresentation of ribosomal genes at embryonic stages in TEC development (Fig. 1b, c). Genes involved in ribosomal biogenesis (KEGG Ribosome) displayed diminishing expression with age in cTEC and mTEC from the high levels observed in the E13.5 TEC and E15.5 cTEC and mTEC (Fig. 2b).

Approximately 85% of RNA in a cell is ribosomal (r)RNA[14]; thus, measuring total RNA abundance is a simple estimate of cellular ribosome content. To examine if the age-dependent decrease in ribosomal protein gene transcripts correlates with a decline in ribosomal concentration, we measured the total RNA abundance of TEC across this data set using Pyronin Y (PY) that intercalates into RNA. Co-staining for intracellular PY and DAPI permitted the exclusion of proliferating DAPI[+] TEC from the analysis, as proliferating cells typically contain higher levels of RNA and TEC proliferation rates vary with age (Fig. 2a). Assessment of PY fluorescence on E17.5, newborn, and adult samples demonstrated age-dependent reduction in total RNA abundance in non-cycling TEC over these developmental timepoints (Fig. 2c).

**Single-cell transcriptome analysis of fetal and adult TEC**. To assess in greater depth the age-dependent transcriptional diversity in TEC, we transcriptionally profiled TEC from different ages using single-cell RNA-seq (scRNA-seq). We focused on cTEC as TEC progenitors have been reported to express cTEC markers prior to differentiating into their downstream cTEC/mTEC

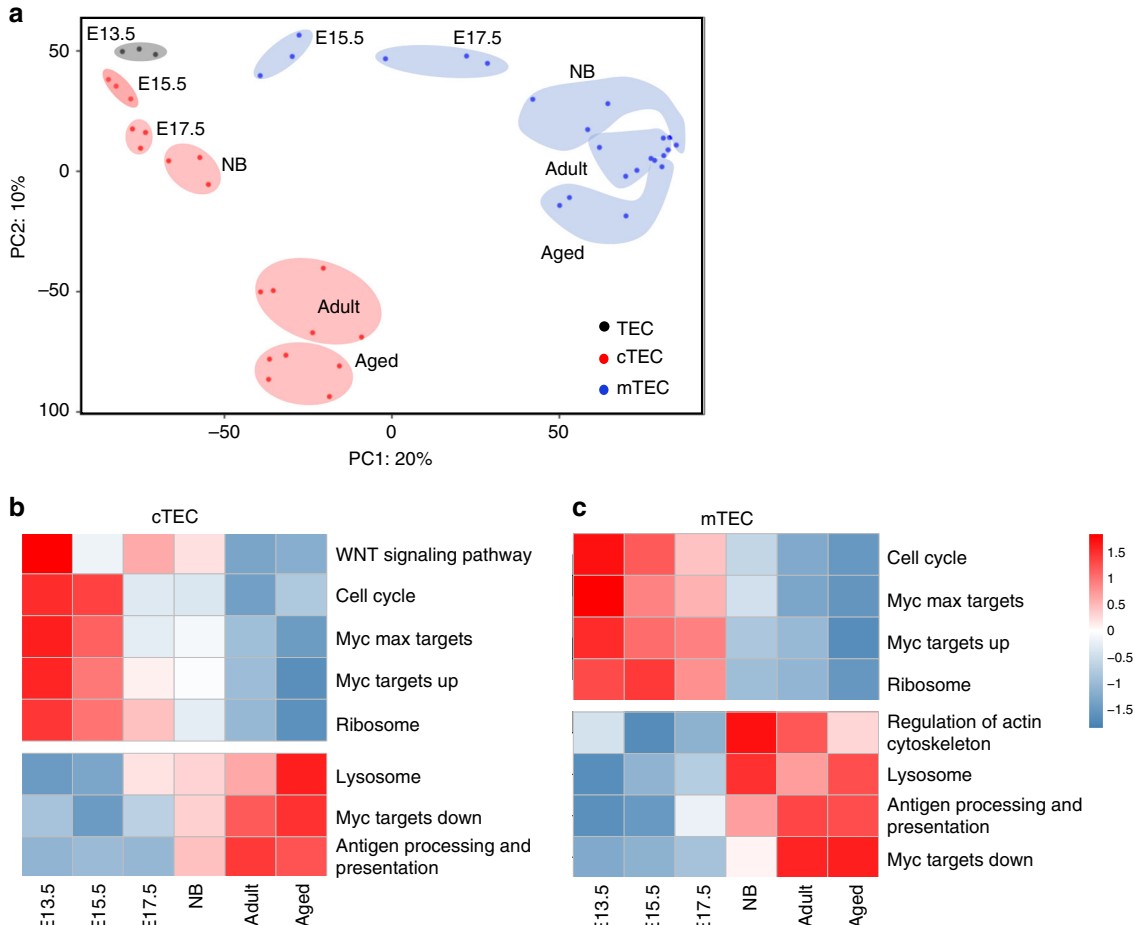

**Fig. 1** Transcriptional diversity of TEC through development. Total TEC (CD45−EpCAM+) from embryonic day 13.5 (E13.5, black circles), or cTEC (CD45−EpCAM+Ly51+UEA−) (red circles) and mTEC (CD45−EpCAM+Ly51−UEA+) (blue circles) from mice at the indicated ages (E15.5, E17.5, newborn (NB), adult or aged were cell sorted for bulk RNA-seq analysis. **a** Two-dimensional representation of populations via a principle component analysis, where each data point is an individual sample and the shaded ovals identify replicate populations. **b**, **c** A row normalized heatmap of expression of gene pathways highly enriched in embryonic or adult cTEC (**b**) or mTEC (**c**) samples at indicated age, identified by gene set variation analysis.

progeny in the fetal thymus[15] and we wanted to include such putative TEC progenitors which might control embryonic thymic expansion. Visualization of data using a t-distributed stochastic neighbor embedding (t-SNE) plot identified clusters of TEC as well as contaminating non-TEC (Supplementary Fig. 2a and b). Newborn and adult cTEC contained a T cell contaminated cluster (cluster 10, Supplementary Fig. 2c) likely representing thymic nurse cells, consisting of CD4+ CD8+ thymocytes enclosed within cTEC[16]. Adult mTEC clustered separately from adult cTEC, and lacked detectable expression of cTEC specific genes including *Psbmb11* (Supplementary Fig. 2d)[17], confirming the purity of the limited number of adult cTEC. In addition, we identified a small subset of described putative TEC progenitors in postnatal mice expressing Plet1, Cldn3, and Cldn4 genes (cluster 13)[18,19]. Contaminating clusters (10, 11, and 12) and adult mTEC were removed before further analysis and all used populations are displayed in the t-SNE plot in Fig. 3a.

**Myc activity declines in TEC during development.** Cell cycle genes can dominate transcriptional variation within cell populations[20]. Therefore, some, if not all distinct gene signatures identified by the bulk RNA-seq in embryonic TEC could conceivably result from high proliferation rates of TEC at early developmental timepoints. We therefore used the scRNA-seq data set to examine

if the observed decline of ribosomal protein transcripts during development occurred within non-proliferating cTEC.

We determined the proliferation status of individual cells in each sample by constructing a proliferation score using the Seurat CellCycleScoring function[21] from a combined list of G2M-phase and S-phase genes[20] (Fig. 3b). The frequency of cells with a high proliferation score declined from E13.5 to newborn and further in adult, consistent with the declining frequency of BrdU+ cTEC with age observed in Fig. 2a. Next, we generated a ribosomal score from the KEGG Ribosome gene set. We observed a dramatic decrease in the ribosomal score between E13.5 and newborn subsets (Fig. 3c); and a further but smaller decline between newborn and adult cTEC, consistent with gene expression profiles at the population level (Fig. 2b). The cTEC were then divided into subsets with low or high proliferation scores (Fig. 3d). Both subsets displayed an equivalent reduction in ribosomal score between E13.5 and newborn, thus indicating that the observed decline in expression of genes implicated in ribosomal biogenesis with age is not a trivial consequence of reduced numbers of cycling cells but is independent of cell cycle status. We additionally tested the strength of the relationship between ribosomal expression and age by fitting a general linear model to the data. Ribosomal score and developmental timepoint were strongly correlated ($R^2 = 0.67$), and remained strongly correlated when cell cycle status was regressed out ($R^2 = 0.67$).

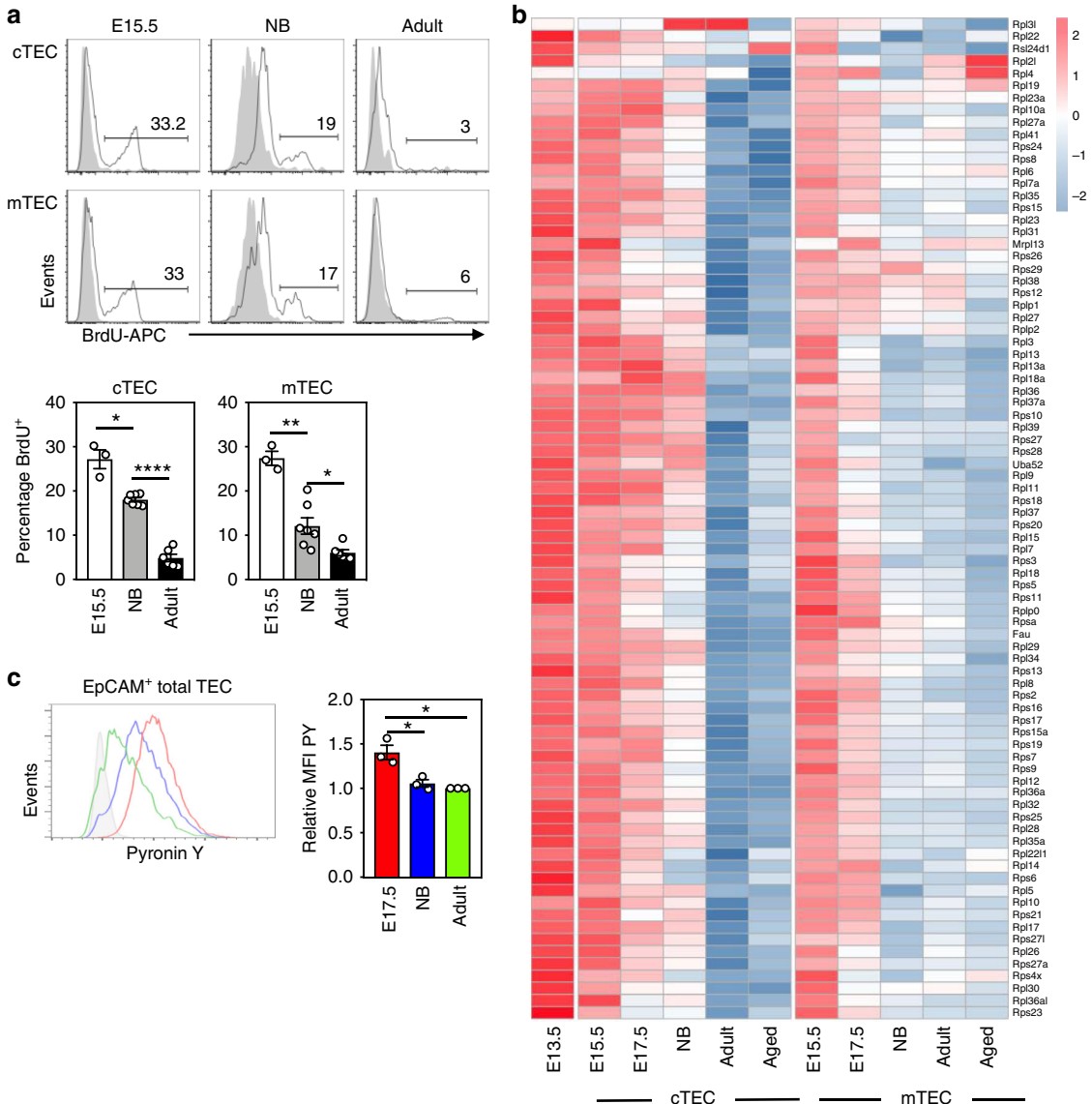

**Fig. 2** TEC have reduced rates of proliferation and ribosomal biogenesis with age. **a** Representative flow plots of the frequency of BrdU⁺ cells in cTEC (CD45⁻EpCAM⁺Ly51⁺UEA⁻) (above) and mTEC (CD45⁻EpCAM⁺Ly51⁻UEA⁺) (below) displayed as histograms (black line) overlaid on the negative control (gray bar). Bar graphs represent the frequencies of BrdU⁺ cTEC and mTEC populations. **b** A row normalized heatmap of expression of KEGG ribosomal genes at indicated ages, where each column represents an average of a minimum of three biological replicates. **c** Representative flow plot of Pyronin Y staining on DAPI⁻CD45⁻EpCAM⁺ total TEC from indicated timepoints, overlaid as a histogram, where gray filled histogram is RNAse treated adult cells. Bar graph represents the MFI of Pyronin Y at each indicated timepoint, relative to the Pyronin Y MFI of the adult sample which is set to 1 and so has no error bar. All bar graphs show mean ± SEM for a minimum $n = 3$ mice per age. A two-tailed unpaired Student's $t$ test was performed to determine significance. *$p < 0.05$, **$p < 0.01$, ****$p < 0.0001$. The source data underlying (**a**, **c**), are provided as a Source Data file.

To confirm the decline in expression of ribosomal genes with increased age was not driven by cellular heterogeneity, such as frequencies of undifferentiated cells, we repeated this analysis using clusters defined by transcriptional similarity (Supplementary Fig. 2b). Interestingly, the proliferation score for the cluster analysis displayed dramatic differences between individual clusters at E15.5 and NB stages (Supplementary Fig. 2e), revealing heterogeneity in cTEC based on proliferation status at these developmental stages (genes differentially expressed between clusters of the same developmental timepoint are listed in Supplementary Data 4). However, cTEC clusters at each developmental timepoint had similar ribosomal scores, that declined with developmental age (Supplementary Fig. 2f). This cluster analysis strengthens the finding that the decline in

expression of ribosomal genes during development is independent of cell cycle. We conclude that this decline in expression of ribosomal genes with age is a key feature of the development of thymic epithelial cells.

Transcription factor Myc is a critical controller of ribosomal biogenesis[22]. We, therefore, examined if Myc protein abundance differed in TEC through development. We used a green fluorescent protein (GFP)-c-Myc knock-in mouse, where GFP is a surrogate for Myc protein. Myc-GFP protein levels were highest in TEC at E13.5 and declined in expression until birth (Fig. 3e) in both cTEC and mTEC. Myc-GFP was not detectable above background in adult TEC of either lineage (Fig. 3f). Interestingly, transcript levels of Myc by bulk RNA-seq analysis did not decrease with age (Supplementary Fig. 3a). The disparity

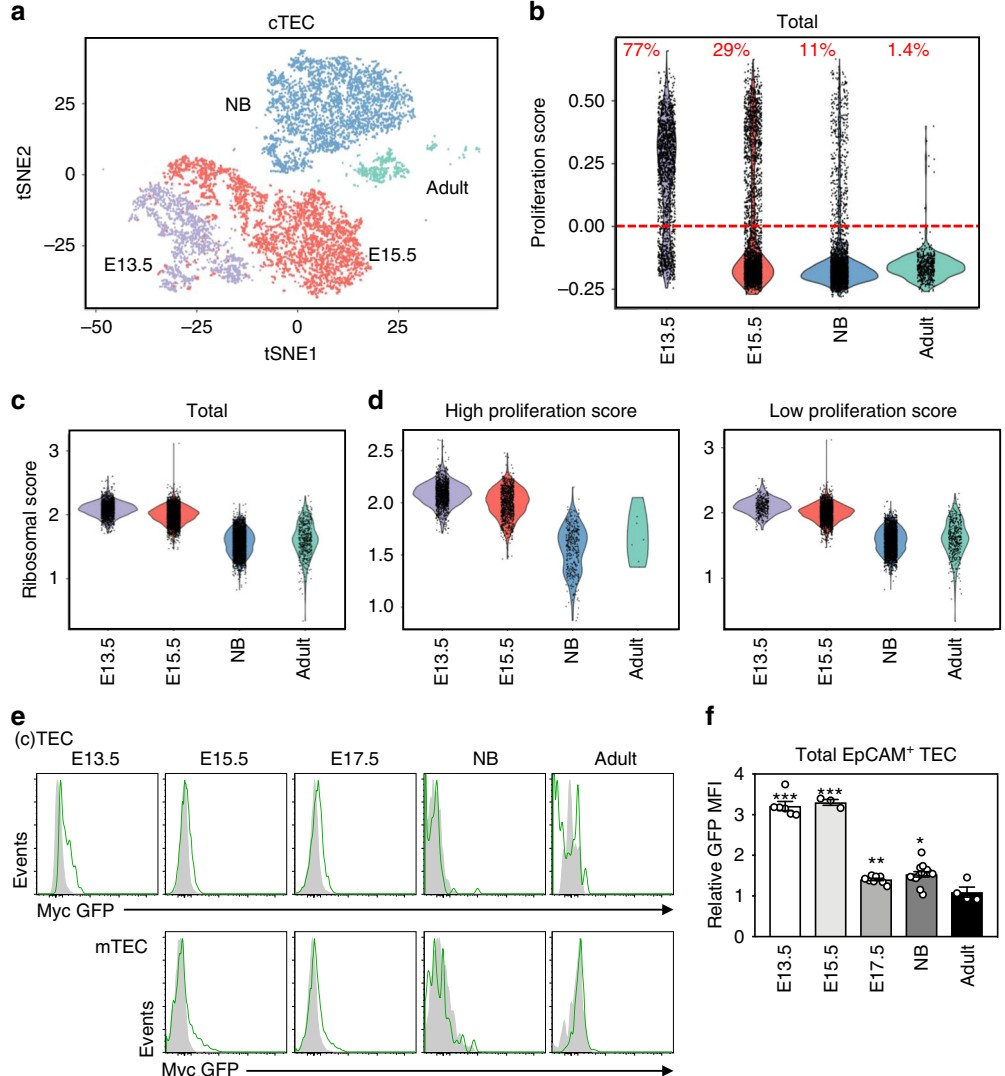

**Fig. 3** A decline in Myc activity and protein levels in TEC during fetal development. Single-cell RNA-seq analysis of cell sorted total TEC (CD45−EpCAM+) from embryonic day 13.5 (E13.5, purple) and cTEC (CD45−EpCAM+Ly51+UEA−) from embryonic day 15.5 (E15.5, pink), newborn (NB, blue) and adult (green) mice. **a** A t-SNE plot of TEC populations, colored and clustered by indicated ages; each dot is a cell. **b** A violin plot of the proliferation score applied to cells at each indicated timepoint. The red line separates cells with a high proliferation score above the line and a low proliferation score below the line; percentages of cells above the line are indicated in red. Violins plots of the ribosomal score applied to total cells (**c**), or cells with a high proliferation and low proliferation score (**d**) at indicated timepoints. **e** Myc-GFP levels displayed as histogram gated on CD45−EpCAM+ total TEC from E13.5, or cTEC (CD45−EpCAM+Ly51+UEA−) (above) and mTEC (CD45−EpCAM+Ly51−UEA+) (below) subsets from homozygote GFP-c-Myc knock-in mice at each indicated age, overlaid on litter mate WT GFP-c-Myc knock-in controls (filled gray histograms). **f** Bar graph represents the relative Myc-GFP MFI against WT aged matched controls in total CD45−EpCAM+ total TEC at each indicated timepoint. Bar graphs show mean ± SEM for a minimum $n = 3$ mice per age. A two-tailed unpaired Student's $t$ test was performed to determine significance. *$p < 0.05$, **$p < 0.01$, ***$p < 0.001$. The source data underlying (**f**) are provided as a Source Data file.

in *Myc* mRNA and protein expression suggests Myc is post-transcriptionally regulated in adult TEC[23]. The results establish that a reduction in Myc protein levels occurs through fetal development, concordant with the earlier observed age-related reduction in expression of Myc target genes in TEC.

**Transgenic expression of Myc in TEC drives thymic growth**. To examine if the decrease in Myc protein observed in TEC after birth limits thymic growth in adult mice, we ectopically expressed Myc in TEC and examined the effects of continual Myc expression on thymic size. We crossed FoxN1Cre recombinase mice[24] to mice with a human *MYC* cDNA transgene inserted into the Rosa-26 locus (R26StopFLMyc)[25]. We called these FoxN1MycTg mice. Population level RNA-seq confirmed increased human

*MYC* mRNA in cTEC, and to a lesser degree in mTEC, in adult FoxN1MycTg mice (Supplementary Fig. 3a). Furthermore, we confirmed an increase in Myc protein by flow cytometry in adult FoxN1MycTg cTEC and mTEC (Supplementary Fig. 3b).

Next, we explored the biological consequences of forced Myc expression in TEC on thymic size. Transgenic expression of Myc in TEC conferred a dramatic increase in thymic size in adult mice (Fig. 4a, b). Whereas transgenic Myc had no effect on thymic size at E14.5 or at the newborn stages in development, by 4 weeks of age mice displayed a twofold increase in thymic size compared with littermate controls (Fig. 4a). The size of the thymus continued to expand into adulthood, causing mice to die from 15 weeks onward[2]. This undetectable increase in size until after birth is similar to other large thymus mouse models[3,4]. It is likely because

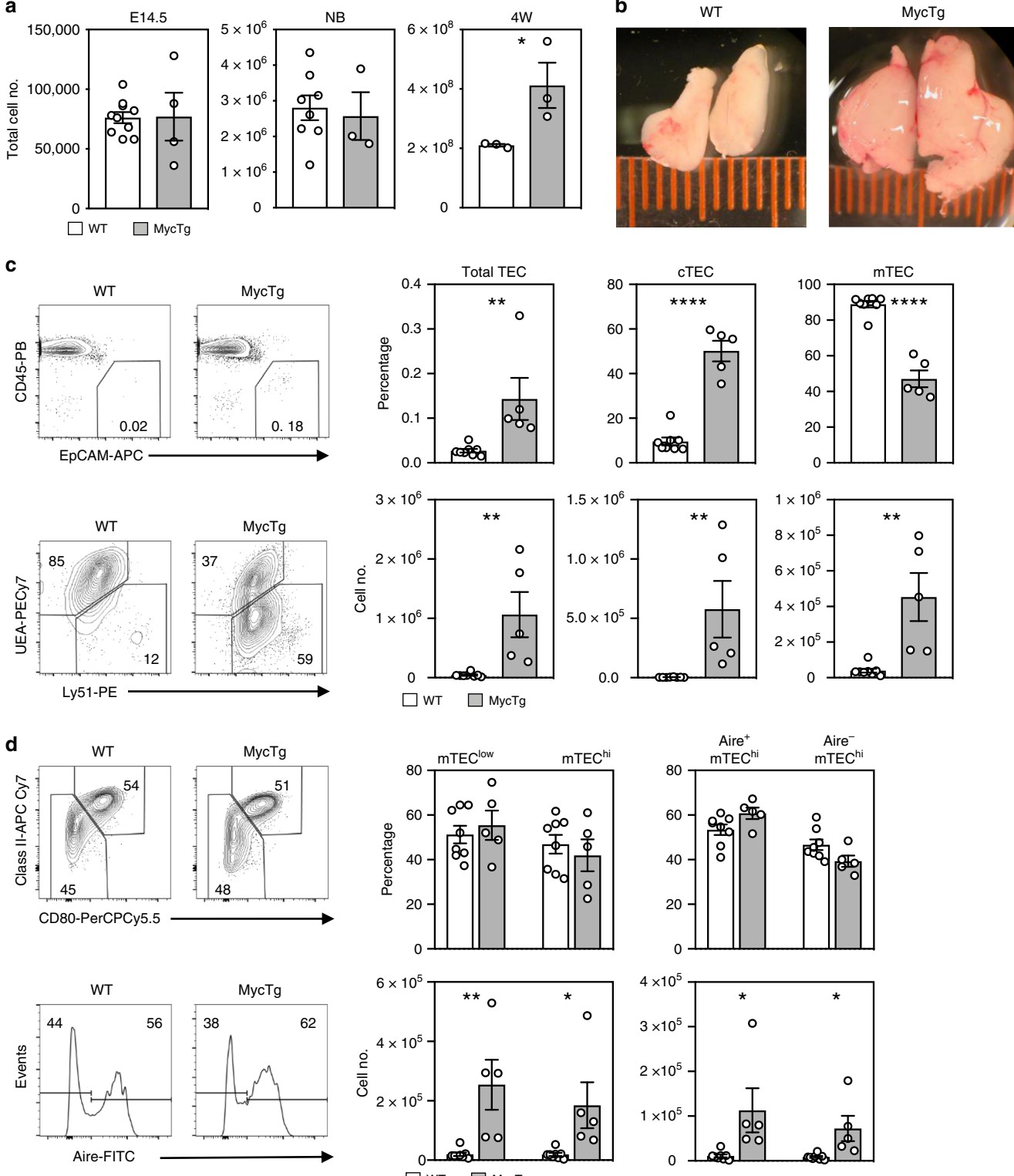

**Fig. 4** Transgenic expression of Myc in TEC drives thymic growth. **a** Total thymus counts of WT (white bars) or FoxN1MycTg (gray bars) mice at embryonic day 14.5 (E14.5), newborn (NB) or 4 weeks of age. **b** Photographs of 10-week-old thymi from WT (left) or FoxN1MycTg (right) mice. **c** Representative flow plots of the frequency of $CD45^-$ $EpCAM^+$ total TEC (above), or $Ly51^+$ $UEA^-$ cTEC and $Ly51^-$ $UEA^+$ mTEC pre-gated on $CD45^-$ $EpCAM^+$ total TEC (below), from 4-week-old WT or FoxN1MycTg mice. Bar graphs display the frequency (above) or cell number (below) of total TEC (left), cTEC (middle) or mTEC (right) of adult WT or FoxN1MycTg mice. **d** Representative flow plots of $CD80^+$ $ClassII^+$ $mTEC^{hi}$ or $CD80^-ClassII^{low}$ $mTEC^{low}$, pre-gated on $Ly51^-$ $UEA^+$ mTEC, or histograms displaying the frequency of $Aire^+$ and $Aire^-$ cells pre-gated on $CD80^+$ $ClassII^+$ $mTEC^{hi}$ from WT or FoxN1MycTg mice. Bar graphs display the frequency (above) or numbers (below) of the corresponding populations. All bar graphs show mean ± SEM for a minimum $n = 3$ mice per genotype. A two-tailed unpaired Student's $t$ test was performed to determine significance. $*p < 0.05$, $**p < 0.01$, $****p < 0.0001$. The source data underlying (**a**), (**c**), (**d**) are provided as a Source Data file.

endogenous Myc already supports high Myc activity during fetal development, and so ectopically expressed Myc has little effect until endogenous Myc has diminished. The frequency and number of EpCAM$^+$ TEC were greatly increased in the thymus of adult FoxN1MycTg mice (Fig. 4c). The ratio of cTEC to mTEC was also altered, with an increased frequency of cTEC recovered from adult FoxN1MycTg thymi; however, the numbers of both TEC lineages were significantly increased in the FoxN1MycTg, due to the dramatic increase in number of TEC (Fig. 4c). FoxN1MycTg mice possessed normal frequencies of CD80$^+$Class II$^+$mTEC$^{hi}$ and CD80$^-$Class II$^{low}$mTEC$^{low}$ subsets, and unaffected frequencies of Aire$^+$mTEC$^{hi}$ (Fig. 4d); again, the numbers of all mTEC subsets were significantly increased in FoxN1MycTg thymi, due to the increase in total numbers. Although the overall size of thymic lobes was considerably larger in FoxN1MycTg mice, thymic architecture remained relatively unchanged, with no obvious defects observed by H&E staining (Supplementary Fig. 4a).

**Transgenic Myc expression in TEC increases thymic function**. The expansion of TEC numbers in FoxN1MycTg mice fostered a dramatic increase in the number of thymocytes recovered from adult thymi. Proportions of the major thymocyte populations were minimally altered in the enlarged thymi, based on CD4, CD8, and TCRβ expression, whilst numbers of each population were increased (Supplementary Fig. 4b and c). More detailed analysis detected some differences within the FoxN1MycTg single positive (SP) CD4 and SP CD8 thymocyte compartments. FoxN1MycTg thymi had a small but consistent increase in frequency of FoxP3$^+$ CD25$^+$ T regulatory cells compared with WT controls (Supplementary Fig. 4d). Furthermore, an increase in the frequency of the most mature CD69$^-$CD62L$^+$ SP4 and SP8 thymocytes was detected within the FoxN1MycTg adult thymus (Supplementary Fig. 4e). However, this was not a consequence of increased proliferation rate (Supplementary Fig. 4f).

The expansion in the TEC compartment increased the number of T cells in the periphery of adult FoxN1MycTg mice[26]. The cellularity of the spleen was unchanged in adult FoxN1MycTg mice, and the frequency and number of extrathymically generated B cells was not significantly altered (Supplementary Fig. 5a). However, increased frequencies and numbers of T cells were measured in the spleen (Supplementary Fig. 5b) due to a significant increase in the number of naive T cells (Supplementary Fig. 5c and d). The numbers of central memory and effector memory CD4 T cells were unaltered in the spleen of the FoxN1MycTg mice (Supplementary Fig. 5c). The numbers of central memory, but not effector memory, CD8 T cells were increased (Supplementary Fig. 5d). In addition, the frequencies of T Regulatory cells were significantly reduced within the periphery of the FoxN1MycTg mice, although the overall increase in the numbers of total CD4 T cells resulted in no alterations to the numbers of this subset (Supplementary Fig. 5e).

The elevated numbers of naive T cells suggested an increase in recent thymic emigrants. We crossed the FoxN1MycTg mice to Rag2-GFP reporter mice, where retained GFP expression marks recent thymic emigrants[27] and confirmed FoxN1MycTgRag2-GFP mice had increased frequencies of Rag2-GFP$^+$ T cells (Supplementary Fig. 5f).

Collectively, the results confirm transgenic expression of Myc in TEC resulted in increased thymus size in adult mice, which consequently increased T cell output.

**Transcriptional effects of transgenic Myc expression**. To establish if the Myc induced TEC expansion and thymic growth in adult FoxN1MycTg mice was mediated by similar mechanisms to those driving thymic growth in the fetal thymus, we examined

the transcriptional effects of enforced Myc expression on TEC. We first determined the changes between adult FoxN1MycTg cTEC and mTEC and WT controls using bulk RNA-seq. This comparison revealed thousands of genes significantly upregulated and downregulated in the FoxN1MycTg TEC compared with WT (Supplementary Fig. 6a), with overlap between the two TEC lineages. Genes upregulated in the FoxN1MycTg cTEC and mTEC were associated with ribosomal biogenesis and translation; whereas genes declining in expression were implicated in antigen processing and presentation, cell adhesion, interferon gamma response, and response to cytokine (Supplementary Fig. 6b and Supplementary Data 5). Interestingly, many of the pathways upregulated in FoxN1MycTg cTEC and mTEC correspond with pathways enriched in WT TEC with age shown in Fig. 1b, c, most strikingly pathways involving the ribosome.

Next, we transcriptionally profiled adult WT and FoxN1MycTg cTEC and mTEC, plus E17.5 cTEC, by scRNA-seq (Fig. 5a and Supplementary Fig. 6c). We removed contaminants (clusters 11 and 14, Supplementary Fig. 6c and d). FoxN1MycTg cTEC and mTEC clustered close to their WT counterparts and did not express fetal-specific genes such as Hmga2 (Supplementary Fig. 6c and d). FoxN1MycTg cTEC separated into two clusters (clusters 3 and 8, Supplementary Fig. 6c), one clustered with the WT cTEC, whilst the other clustered independently. Genes differently expressed between these two clusters are listed in Supplementary Data 6. FoxN1MycTg cTEC and mTEC displayed an increased ribosomal score compared with adult WT cTEC and mTEC (Fig. 5c). The proliferation score was also increased in the FoxN1MycTg cTEC and mTEC compared with adult WT (Fig. 5b).

Collectively, these results suggest the transcriptional effects of the Myc transgene in TEC results in a dramatic increase in ribosomal biogenesis and a more modest rise in rates of proliferation. Thus, the transcriptional consequences of forced Myc expression in adult FoxN1MycTg mice closely mimicked those transcriptional patterns that indicated high Myc activity in early life in TEC.

Nonetheless, adult FoxN1MycTg cTEC did not cluster with the E17.5 cTEC, because some fetal genes were not rescued by transgenic Myc expression, indicating that transgenic Myc represented only one component of the fetal transcriptional program as it only conferred expression of a limited set of fetal enriched genes. We generated a list of fetal genes that were highly expressed in WT fetal samples, and not upregulated or only minimally upregulated in adult FoxN1MycTg cTEC (Supplementary Fig. 7a). This list was used to generate a fetal score and applied to the scRNA-seq data set in Fig. 5 (Supplementary Fig. 7b). This assessment confirmed that these index genes are not highly expressed in any adult WT cTEC or mTEC and are not upregulated in FoxN1MycTg TEC. We speculate this gene list contains upstream Myc controllers.

We explored the transcriptional changes of TEC isolated from another large thymic mouse model. RNA sequencing of mTEC from adult Keratin 5 promoter-driven cyclin D1-transgenic (K5CyclinD1) mice[4] confirmed an increase in expression of cell cycle genes (Supplementary Fig. 7c)[28], but interestingly, without an increase in expression of ribosomal genes (Supplementary Fig. 7d). Thus, Myc induced thymic growth is distinct from growth resulting from forced expression of the Cyclin D1 cell cycle controller.

**Transgenic Myc expression in TEC does not alter mTEC subsets**. We next examined the effects of the Myc transgene on TEC diversity and heterogeneity. As cTEC subsets are not well described, we focused the analysis on the four major mTEC

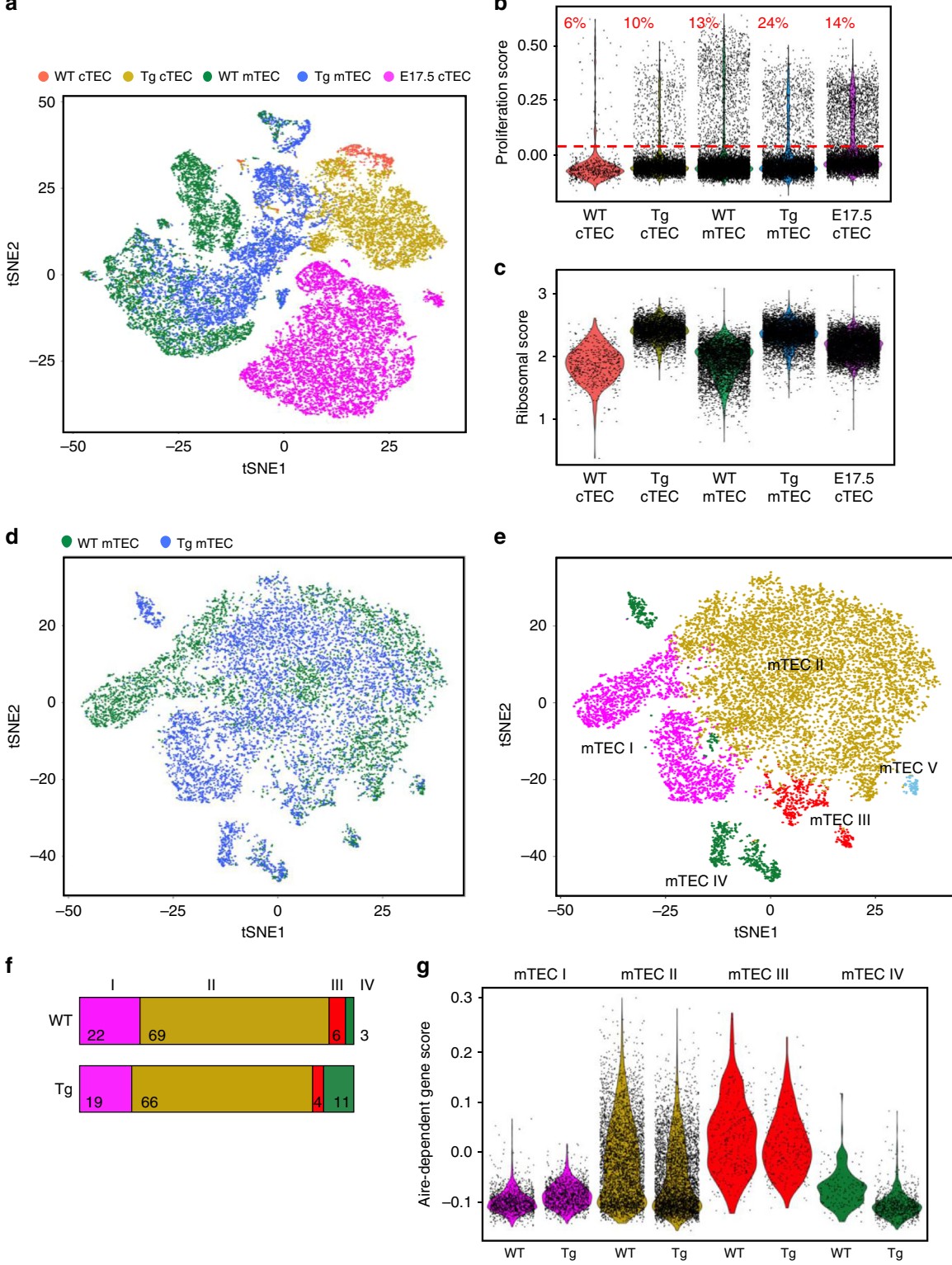

**Fig. 5** Transgenic Myc confers expression of a limited set of fetal-specific genes in adult TEC. scRNA-seq analysis performed on cTEC (CD45−EpCAM+ Ly51+UEA−) or mTEC (CD45−EpCAM+Ly51−UEA+) isolated from adult WT or FoxN1MycTg mice, and a cTEC sample from embryonic day 17.5 (E17.5 cTEC). **a** A t-SNE representation of the populations, colored and clustered by original identity. **b** A violin plot of the proliferation score for the indicated populations. The red line separates cells based on their proliferation score, high score above the line and low score below the line, percentages of cells above the line are indicated in red. **c** A violin plot of the ribosomal score applied to the indicated populations. **d**, **e** A t-SNE representation, regressed out for cell cycle, of WT and FoxN1MycTg mTEC populations, colored and clustered by original identity (**d**) or gene expression similarity (**e**). **f** The fraction of four (of the five) mTEC clusters identified in (**e**), for WT mTEC (above) or FoxN1MycTg mTEC (below) samples. The colors of the bars correspond to colors allocated to the four clusters in the t-SNE plot displayed in (**e**). **g** A violin plot of the Aire-dependent gene score of either the WT or FoxN1MycTg cells that make up each of the four indicated mTEC subsets.

subsets, that have been newly defined by scRNA-seq[11], and examined if FoxN1MycTg mTEC had altered sub-population structure (Fig. 5d). We performed cluster analysis on cell cycle regressed data, to reduce differences driven by altered proliferation rates between WT and FoxN1MycTg, at a resolution that enabled the identification of the four mTEC subsets (Fig. 5e), based on expression of selected marker genes (Supplementary Fig. 8a)[11]. All four mTEC clusters contained both WT and FoxN1MycTg cells (Fig. 5d, f), in addition to a small unidentified cluster (mTECV, Fig. 5e) that we did not further consider at this time. The most highly differentiated genes per cluster are in Supplementary Data 7. The proportions of mTEC subsets were relatively unchanged between the WT and FoxN1MycTg (Fig. 5f), except for an increase in mTEC IV thymic tuft cells in the FoxN1MycTg sample. The most highly differentiated genes in the WT versus FoxN1MycTg cells per cluster are listed in Supplementary Data 8. The preserved frequency of Aire-expressing cells was complementary to flow cytometric data (Fig. 4d). All FoxN1MycTg cells in the four mTEC clusters had an increased ribosomal score compared with the WT cells of that cluster (Supplementary Fig. 8c) and an increased proliferation score in three of the mTEC clusters (Supplementary Fig. 8b). We additionally examined the expression of known Aire-dependent genes in each cluster, for the WT or FoxN1MycTg samples (Fig. 5g)[11]. All clusters displayed comparable levels of expression of Aire-dependent genes.

**Myc expression increases the frequency of proliferating TEC**. To determine whether ectopic Myc expression increased rates of proliferation, we performed in vivo single pulse BrdU labeling. This identified a twofold increase in the frequency of BrdU$^+$ proliferating adult FoxN1MycTg TEC (Fig. 6a). As an control, CD45$^+$ thymocytes in the same samples did not display an increased rate of proliferation. This result confirmed that conditional ectopic Myc expression forced increased rates of proliferation in adult TEC.

**Myc expression increases numbers of active ribosomes**. To confirm the Myc transgene increased ribosomes in adult TEC, we measured total RNA abundance in the FoxN1MycTg TEC. The FoxN1MycTg TEC had increased PY MFI compared with controls (Fig. 6b), and thus, higher total RNA abundance per cell. Moreover, we assessed alterations to the amount of active protein synthesis in adult TEC with ectopic Myc expression using ribopuromycylation (RPM)[29]. We detected significantly more RPM staining in adult FoxN1MycTg TEC (Fig. 6c), and therefore an increased number of actively translating ribosomes.

These results demonstrated that transgenic expression of Myc in TEC resulted in increased rates of ribosomal biogenesis and proliferation in the adult thymus that in turn promoted thymic growth. This confirms that cell proliferation and ribosomal biogenesis are downstream targets of Myc in TEC. The high abundance of Myc protein in fetal TEC, in addition to the high rates of cell proliferation and ribosomal biogenesis at this developmental timepoint, indicates Myc is a regulator of the transcriptional diversity between fetal and adult TEC, that supports the rapid expansion of thymic size during embryogenesis.

**Myc expression in adult TEC confers fetal properties**. The engraftment potential of TEC following intrathymic transfer is lost by one month of age[30]. We, therefore, assessed if ectopic Myc expression in adult TEC conferred transplantability. TEC enriched thymic samples were prepared from adult FoxN1MycTg or WT controls and intrathymically injected into sub-lethally

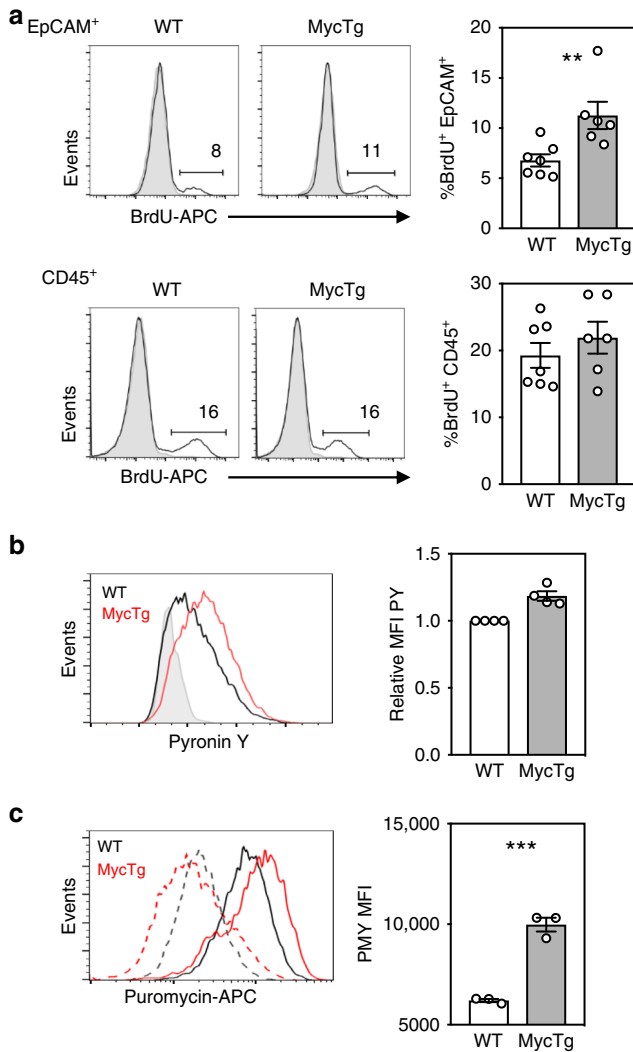

**Fig. 6** Myc expression increases the frequency of proliferating TEC and the number of actively translating ribosomes. **a** Representative histograms and bar graphs of the frequency of BrdU$^+$ CD45$^-$ EpCAM$^+$ total TEC (above) or BrdU$^+$ CD45$^+$ EpCAM$^-$ total thymocytes (below) in adult WT or FoxN1MycTg mice. Gray lines on histograms represent the negative control. **b** A histogram of pyronin Y staining on CD45$^-$EpCAM$^+$ total TEC populations from WT adult (black line) or FoxN1MycTg adults (red line), overlaid against an adult WT sample treated with RNAse (gray line). Bar graph represents pyronin Y MFI relative to the adult WT control, which is set to 1 in all experiments and so has no error bar. **c** A histogram of puromycin antibody staining on CD45$^-$ EpCAM$^+$ total TEC from adult WT (black) or FoxN1MycTg (red) mice treated with either puromycin (PMY) plus cxcycloheximide (CHX) staining (solid lines) or puromycin plus harringtonine (HNT) (dotted lines). Bar graph represents the MFI of puromycin (PMY) gated on total TEC from WT adult or FoxN1MycTg adults, calculated from the MFI of PMY + CHX treated mice, from which we subtracted the MFI from PMY + HNT treated mice to eliminate the signal associated with stalled ribosomes. All bar graphs show mean ± SEM for a minimum of $n = 3$ mice per genotype. A two-tailed unpaired Student's $t$ test was performed to determine significance. $**p < 0.01$, $***p < 0.001$. The source data underlying figures are provided as a Source Data file.

irradiated WT recipients (Fig. 7a). The percentage of EpCAM$^+$ TEC recovered from hosts receiving FoxN1MycTg donor cells was higher than hosts receiving WT donor cells (Fig. 7b). The thymus counts from hosts receiving FoxN1MycTg donor cells were also significantly increased, resulting in a higher number of

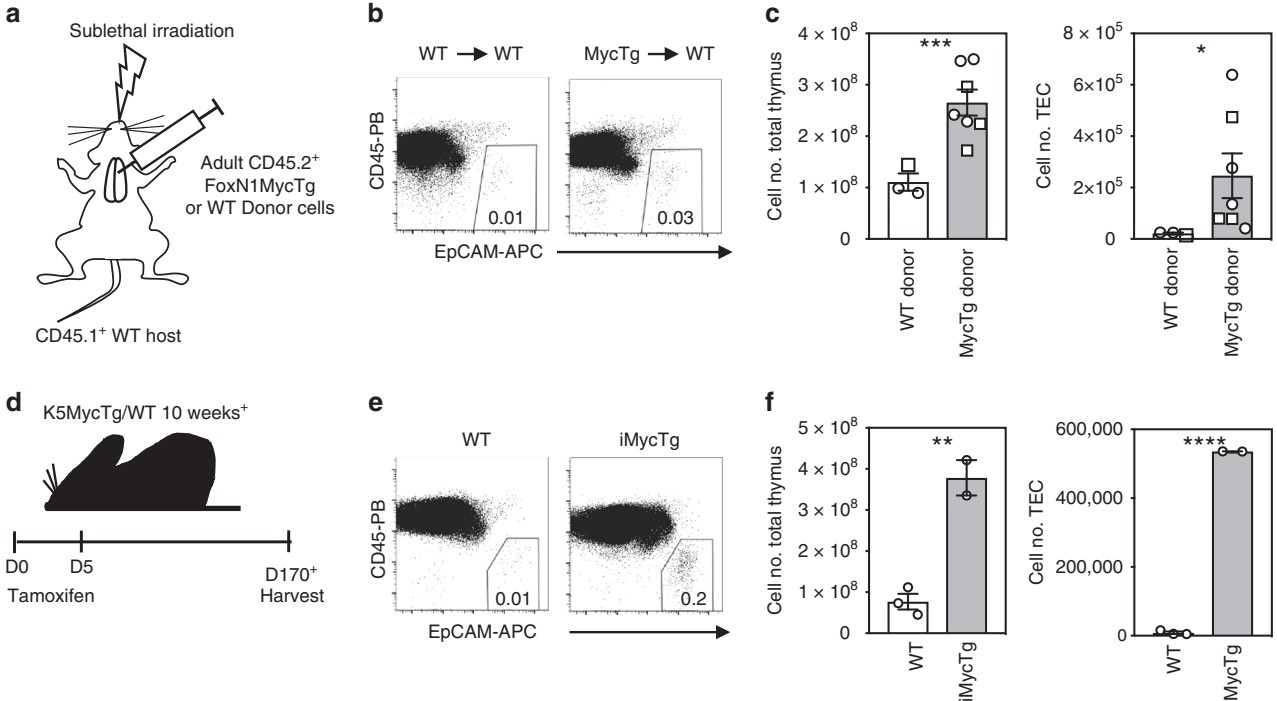

**Fig. 7** Transgenic Myc expression in adult TEC confers fetal properties and can drive thymic regeneration. **a** Schematic of the experimental design of intrathymic injections. **b** Representative flow plots of the frequency of CD45−EpCAM+ total TEC recovered from WT hosts receiving adult WT or FoxN1MycTg donor TEC. **c** Bar graphs display total cells in thymus (left) and CD45−EpCAM+ total TEC counts (right) from WT host thymi receiving WT (white) or FoxN1MycTg (gray) donor TEC. Bar graphs show mean ± SEM, where circles display experiment 1 and squares display experiment 2. **d** A schematic of the experimental design of tamoxifen induced Myc expression in TEC. **e** Representative flow plots of the frequency of CD45−EpCAM+ total TEC recovered from K5−MycTg+ (WT) or K5+MycTg+ (iMycTg) tamoxifen treated mice. **f** Bar graphs display total thymus counts or CD45−EpCAM+ total TEC counts from K5−MycTg+ (white) or K5+MycTg+ (gray stripe) tamoxifen treated mice. Bar graphs show mean ± SEM for a minimum $n = 2$ mice per genotype. A two-tailed unpaired Student's $t$ test was performed to determine significance. $*p < 0.05$, $**p < 0.01$, $****p < 0.0001$. The source data underlying (**c**, **f**) are provided as a Source Data file.

TEC per thymus (Fig. 7c). Hence, continued transgenic expression of Myc into adulthood in TEC preserves their engraftment ability. Moreover, transplantation of adult FoxN1MycTg TEC into WT adult thymi drives thymic growth.

**Myc expression in adult TEC drives thymic regeneration**. We next assessed if re-expressing Myc in adult TEC was sufficient to stimulate thymic growth, even after the fetal program had been extinguished. We generated an inducible MycTg mouse model, by crossing the floxed MycTg mouse to a K5-CreER[T2]Cre mouse[31], named here iMycTg. Administration of tamoxifen activates the ER[T2]-Cre controlled by Keratin5 locus control elements and results in the expression of the Myc transgene. Keratin 5 is expressed in TEC progenitor subsets and mTEC, therefore, this model facilitated restricted expression of the Myc transgene in TEC after tamoxifen administration[26].

After 5 days of tamoxifen treatment, adult K5+MycTg+ and K5−MycTg+ mice were left for an additional 60 days+ before sacrificing and determining TEC and thymic cell counts (Fig. 7d). The percentage of EpCAM+ TEC recovered from tamoxifen treated iMycTg mice was tenfold higher than WT controls (Fig. 7e), similar to the percentages recovered from the FoxN1MycTg mice (Fig. 4c). Furthermore, the thymus counts were significantly increased in iMycTg mice, with a fourfold rise in cell numbers (Fig. 7f). This resulted in a highly significant expansion of TEC numbers in the iMycTg thymi. Therefore, the Myc transgene does not need to be continually expressed from fetal stages in development to drive thymic growth, but instead,

re-expression of Myc in adult TEC is sufficient to promote thymic growth and regeneration.

## Discussion

Thymic epithelial cells are essential in generating a functional, diverse and self-tolerant T cell repertoire[8]. Here, we used population and single-cell RNA-seq to identify distinct transcriptional programs of TEC at fetal and adult stages. We discovered increased expression of gene targets of Myc at fetal stages in development, including genes controlling ribosomal biogenesis and the cell cycle. We found a stepwise decline in expression of these genes through fetal development, that was concordant with declining expression of Myc protein. By conditionally enhancing Myc protein expression in TEC we demonstrated that Myc conferred expression of a limited fetal transcriptional program, that included genes involved in cell cycle progression and ribosomal biogenesis; and enhanced thymic growth from early adulthood without jeopardizing thymic function. Moreover, enforced Myc expression preserved the fetal-specific engraftment potential of TEC into adulthood, and drove thymic growth in the adult or aging thymus. These results establish that a decline in Myc activity from fetal through adult life controls the rate of thymus growth and limits overall thymus size. Collectively, our findings establish two distinct age-related TEC transcriptional programs: rapid growth at early developmental times leading to maintenance of a functioning thymus in adulthood.

While Myc is known to control organ size in *Drosophila*[5] its contribution in vertebrates was previously unclear. Here, we demonstrate that enhanced Myc activity in TEC drives organ

growth in mice. How Myc mediates its role in different cellular processes has been intensely studied, because dysregulation of Myc activity is connected to the development of a large range of human cancers[32,33]. Myc has been hypothesized to be a universal amplifier of all active genes, offering an explanation as to how it is involved in so many biological processes[34]. In our study, TEC-specific transgenic expression of Myc allowed the identification of genes whose relative expression increased in response to increased Myc activity in TEC. These included genes involved in cell cycle, and more strikingly, ribosomal biogenesis, which is rate-limiting for cell growth and replication[35]. We confirmed that ectopic expression of Myc increased the number of actively translating ribosomes, together with the abundance of total RNA in TEC, and elevated the fraction of proliferating TEC. Furthermore, K5CyclinD1 TEC transcriptional profiling confirmed that increased cell cycle was not sufficient to cause an increase in expression of ribosomal protein genes[2], indicating that the changes we saw were not merely consequences of increased rates of cell cycle in fetal TEC.

Although the major focus of this transcriptome analysis was on changes that occur between fetal stages to young adulthood, further declines in Myc activity were evident from adult to aged cTEC. Thus, the dramatic reduction in Myc activity we observed during fetal development continues to occur at more modest rates from adulthood to aging, and we speculate this underlies the reduction in thymic size during involution. It follows that manipulations that upregulate Myc activity could prevent or reverse thymic involution and perhaps enhance thymus regeneration. This conclusion is further strengthened by our observation that intrathymic transplantation of adult FoxN1MycTg TEC can promote thymic growth, and that inducing Myc expression in the adult or aged thymus can also support dramatic increases in thymic size. Hence, physiological regulators of Myc are candidates for therapeutic interventions to improve thymic function.

Transcriptional profiling identified many genes with high levels of expression in fetal TEC that are not upregulated in adult FoxN1MycTg TEC, indicating they are not downstream of Myc. This list of fetal-specific genes may control fetal-specific TEC properties critical for thymic function that are independent of Myc. Such genes, together with Myc and its many targets, constitute a fetal transcriptional program that underlies the exclusive properties of fetal TEC. In addition, they offer a starting point to identify upstream controllers of Myc. This gene list includes Igf2, Igf2bp3, and Igf2bp1. Igf family members have known roles in controlling Myc expression[36], and the RNA-binding proteins Igf2bp1 and Igf2bp3 bind *Myc* mRNA[37,38]. Igf2bp1 promotes *Myc* stability by preventing its mRNA degradation via its association with a sequence in the *Myc* open reading frame, termed the coding region determinant (CRD)[37]. Igf2bp1/2/3 enhance mRNA translation via recognition of m[6]A in mRNA[39], and *Myc* is established to possess m[6]A in mouse HSCs[40], suggesting a mechanism by which Igf2bp RNA-binding proteins might control *Myc* mRNA stability and translation.

Although postnatal TEC expressed *Myc* mRNA, they lacked Myc protein. This discrepancy between Myc protein and *Myc* transcripts levels indicates that Myc is post-transcriptionally and/or post-translationally regulated in postnatal TEC. *Myc* mRNA stability is controlled by different mechanisms[41]. Multiple miR-NAs target Myc[42,43]. In addition, post-translational modifications control Myc activity and function[44]. Interestingly, and distinct from non-transgenic adult TEC, FoxN1MycTg adult TEC clearly expressed Myc protein and exhibited increased Myc activity. The FoxN1MycTg mouse has a human MYC cDNA that lacks the 5′ and 3′ UTR as well as introns[25]; these may be the targets of post-transcriptional control in adult TEC. In addition, human and mouse Myc are 89% conserved at the amino acid level (NCBI

Protein database), and some of the differing amino acids may be the site of post-translational modifications that affect protein stability.

In summary, our findings reveal dramatic transcriptional differences between TEC from the embryonic and adult thymus, providing mechanistic insight into transcriptional and functional differences between fetal and adult TEC. By clearly demonstrating that Myc controls thymic growth and size, we establish Myc as a possible therapeutic target in modulating thymic function and regeneration.

## Methods

**Mice**. B6 (CD45.2) and B6-Ly5.1 (CD45.1) mice were purchased from Jackson Laboratory. The MycTg mice[25], in which a loxP-flanked 'stop' cassette precedes a human MYC cDNA inserted into the *Rosa-26* locus, were purchased from Jackson Laboratory (Stock No: 020458) and crossed with FoxN1Cre mice that were a gift from G.Hollander[24]. Myc-GFP mice[45], with Myc fused to GFP in the N-terminus, were purchased from Jackson Laboratory (Stock No: 21935).

Krt5-CreERT2 knock-in mice[31] were purchased from Jackson Laboratory (Stock No: 029155). Rag2-EGFP mice[46] were purchased from Jackson Laboratory (Stock No: 005688). Mice described as newborn were 0–2 days in age and of either sex. Mice described as adult for flow cytometry/sequencing experiments were 4–6 weeks of age and of either sex (unless otherwise stated). Aged mice were between 18 and 21 months of age. The ages of embryonic mice are specified with E0.5 being noon of the day of the discovered plug. Animal procedures were approved by relevant National Institutes of Health Animal Care and Use Committees.

**Tissue preparation**. Thymus and spleen were dissected into RPMI (ThermoFisher Scientific) containing 5% newborn calf serum (NCS) (Atlanta Biologicals) and mechanically teased with forceps to prepare a single-cell suspension. Splenocytes were treated with ACK Lysing Buffer (Lonza) for 3 min on ice to eliminate red blood cells. When analyzing newborn and adult thymic epithelial cells, single-cell thymic suspensions were enzymatically digested with Liberase TM (63 µg/ml; Roche) and DNase I (20 µg/ml; Roche) for 40 min shaking at 37 °C. Following digestion, epithelial cells were enriched by centrifugation in a Percoll gradient. Briefly, cells were resuspended in 4 ml of 1.115 g ml$^{-1}$ isotonic Percoll (GE Healthcare Life Science) and overlaid with 2 ml of 1.065 g/mL isotonic Percoll, followed by a layer of 2 ml of PBS. Samples were then centrifuged at 2700 RPM at 4 °C for 30 min with the brakes off. The TEC enriched samples were taken from the interface between the Percoll and PBS layer. Embryonic thymi were disaggregated in 0.25% trypsin/0.02% EDTA (Sigma) solution at 37 °C for 7–10 min, with a single-cell suspension made by gentle repetitive pipetting. Embryonic thymi were not subject to an enrichment step. All cell suspensions were then further processed and stained as described below.

**Flow cytometry**. Thymocytes and splenocytes were stained and analyzed in FACS buffer (PBS containing 0.1% sodium azide, 1 mM EDTA, and 0.5% BSA). Thymic epithelial cell preps were analyzed in MACS buffer (PBS containing 2 mM EDTA and 0.5% FCS). Antibodies specific for CD45.2 (104), CD45.1(A20) CD4 (GK1.5), CD8-α (53-6.72), TCR-β (H57), B220 (RA3-6B2), CD19 (1D3), Ly51 (6C3), EpCAM (G8.8), CD80 (16-10A1), MHC Class II (M5/114.15.2), CD44 (IM7), CD62L (MEL-14), CD25 (PC61.5), CD69 (H1.2F3), and Streptavidin PECy7 were from eBioscience. Biotinylated UEA-1 (B-1065) was from Vector Labs. LIVE/DEAD discrimination was performed by staining with DAPI (Sigma). All analyses were presented on singlet live cells. For intracellular staining of Myc protein and Aire, cells were first stained for cell surface molecules, permeabilized using the eBioscience's transcription factor staining buffer set (cat: 00-5523-00) according to the manufacturer's instructions and then either stained for c-Myc rabbit monoclonal antibody (Cell Signaling; D84C12), followed by Goat anti Rabbit Alexa Fluor® 488 secondary (Invitrogen; A11008); or Aire (5H12). Samples were acquired using a flow cytometer (LSRFortessa; BD) and analyzed using FlowJo software (Tree Star). TEC were sorted using an Aria flow cytometer (BD). All cell sort purities were >98%. Absolute cell numbers were obtained using an Accuri C6 PLUS flow cytometer (BD). All newborn and adult TEC numbers were calculated from the frequency of CD45- EpCAM + cells of pre-enrichment samples.

**BrdU staining**. Mice were injected intraperitoneally (i.p) with BrdU (1.5 mg) 18–20 h prior to sacrifice, and thymi processed as stated above. Cell suspensions were intracellular stained for BrdU using the BD bioscience APC BrdU flow kit (552598), according to the manufacturer's instructions. Control mice received vehicle alone injections.

**Pyronin Y staining**. All cell suspensions of adult and embryonic thymi were fixed and permeabilized using eBioscience's transcription factor staining buffer set according to the manufacturer's instructions and first stained with DAPI

(25 mg/ml), followed by Pyronin Y (Sigma P9172-1G) (1 μg/ml). Control cells were treated with RNAse A (Millipore 70856) at 10 mg/ml at 37 °C for 1 h.

**Puromycin staining**. The antibiotic puromycin (PMY) is an aminoacyl-tRNA analog that binds to the ribosome A-site and is covalently incorporated onto the 3′ end of the nascent polypeptide resulting in chain termination. After fixation and permeabilization, the puromycylated polypeptides can be detected by an intracellular stain with an antibody specific to puromycin. Treatment with cxcycloheximide (CHX) prior to PMY stabilizes the ribosome-nascent chain complex and prevents puromycylated polypeptide release. As puromycin can also incorporate into stalled ribosomes, i.e, ribosomes that are not actively translating the bound mRNA, control mice were treated with harringtonine (HNT) prior to PMY. HNT inhibits translation initiation, while allowing currently elongating ribosomes to finish translation of the bound mRNA. The intracellular RPM signal from CHX and PMY treated mice minus the RPM signal after HNT and PMY treatment generates a measure of the number of active ribosomes[29]. Mice were injected intravenously with 100 μl of harringtonine solution (HNT) (1 mg/ml, Santa Cruz Biotechnology) or cycloheximide solution (CHX) (3.4 mg/ml, Enzo Life Sciences) for 15 min before intravenous injection with 100 μl of puromycin (PMY) (10 mg/ml, Calbiochem) in PBS (Life Technologies) that was warmed to 37 °C. Mice were sacrificed 10 min post PMY injection and thymi processed as stated above. Samples were stained with LIVE/DEAD® Fixable Blue Dead Cell Stain Kit (ThermoFisher Scientific) on ice for 10 min, then fixed and permeabilized in fixation/permeabilization buffer (1% paraformaldehyde, ThermoFisher Scientific; 0.0075% digitonin, Wako) in PBS for 20 min at 4 °C. Cells were then labeled with a directly conjugated (using Life Technologies protein labeling kit as per the manufacturer's instructions) anti-PMY Ab (clone PMY-2A4) for 1 h.

**H&E staining**. Thymi were harvested from adult FoxN1MycTg mice, fixed in 4% paraformaldehyde (ThermoFisher Scientific), and mounted in paraffin. Five- to eight-micrometer sections were cut and stained with H&E (performed by an NIH core facility). The central sections were imaged using a Leica MZ12₅ microscope and a Nikon Coolpix 5000 camera.

**Bulk RNA sequencing**. A minimum of three replicates of cTEC and mTEC subsets from each age point were isolated by cell sorting from B6 WT mice according to the gating strategy shown in Supplementary Fig. 1a. Three additional cTEC and mTEC samples were isolated from FoxN1MycTg adult mice. FoxN1MycTg adult samples were 10 weeks of age. We did not use Spike-in controls because a standard methodology is not available for their use with small cell numbers (2000 or more cells in these experiments). Replicates were isolated from individual mice in two or more independent experiments (independent cell isolation, RNA extraction, library preparation, and RNA sequencing). RNA was extracted using the RNeasy plus micro kit (Qiagen) according to the manufacturer's instructions. Quality control was performed by Bioanalyzer (Agilent), and RNA samples with an RNA integrity number > 9 were subsequently used. mRNA sequencing libraries were prepared using the SMARTer Ultra Low Input RNA kit v3 (Clontech) and Nextera XT DNA library preparation kit (Illumina). Paired-end sequence reads of 126 bp were generated by a HiSeq2500 sequencer (Illumina). The raw RNA-seq FASTQ reads were aligned to the mouse genome (mm10) using STAR (v. 2.5.2b[47]) on two-pass mode with mouse Gencode (release 15) gene transfer format. Genes were subsequently counted using Rsubread[48] and because we found T cell marker genes in adult cTEC samples specifically, we used k-means clustering to find a gene cluster which was enriched for known T cell markers (Cd247, Rag1, Cd3e, Cd3g, Cd3d, Cd4, Cd28) using the pheatmap package (Kolde 2018). A full list of the 292T cell contaminant genes is provided in Supplementary Data 9. These genes were removed from all samples prior to analysis for gene expression changes (Aged vs Adult, Fetal vs Adult, and WT vs FoxN1MycTg) using limma-voom with quantile normalization PCA plots were generated using prcomp function from stats package. Venn diagrams show intersection of gene lists derived using twofold change and $p < 0.05$. Pathway enrichment analyses for the Adult vs Aged (Supplementary Fig. 1e) and WT vs FoxN1MycTg (Supplementary Fig. 6b) comparisons were subsequently performed using the top 1000 genes (sorted by t-statistic) in each direction using the Reactome, Molecular Signature Database (MSigDB[49,50]): Hallmark, C2, GO, and KEGG pathways, using the Fisher's exact test (https://github.com/CCBR/l2p).

Differentially expressed genes across cTEC (Fig. 1b) and across mTEC groups (Fig. 1c) were determined using the limma package[10], each set including the 13.5 TEC timepoint as the baseline. The top 2000 genes were subjected to Gene Set Variation Analysis (GSVA[9]) to calculate individual gene set enrichment scores using the MSigDB C2:curated and H:hallmark and C5:GO Gene Ontology databases. Visualization was performed using R (R Development CoreTeam[51]). To confirm consistency between biological replicates, we performed unsupervised hierarchical clustering using a minimum spanning tree. All data presented in heat maps are counts per million (CPM), normalized and log2 transformed. Human *MYC* counts were quantified by running STAR aligning to the human genome (hg38) using Gencode v.25 gtf. K5CyclinD1 and WT mTEC populations were isolated and sequenced according to methods stated in Ohigashi et al.[28].

**Single-cell RNA sequencing**. TEC populations were isolated by cell sorting from B6 WT or FoxN1MycTg mice according to the gating strategy shown in Supplementary Fig. 1a and b. We used the droplet-based GemCode Chromium technology to capture the two separate data sets. Data set one in Fig. 3: E13.5 total TEC (1981) E15.5 cTEC (3571) newborn cTEC (3605), adult cTEC (507), adult mTEC (1586), each expressing a median of 1764–2839 genes per sample. Data set two in Fig. 5: E17.5 cTEC (8381 cells) adult cTEC (514 cells), adult mTEC (5428 cells) adult FoxN1MycTg cTEC (4670 cells), adult FoxN1MycTg mTEC (6736 cells), expressing medians of 2500–3200 genes per sample and all captured at the same time. scRNA-seq libraries were prepared using the Chromium Single Cell 3′ kits, according to the manufacturer's instructions (v2 chemistry, 10x Genomics). The obtained libraries were sequenced with a NextSeq (v2 chemistry, Illumina). Primary analysis was performed with the Cellranger v2.0.1 software using the default parameters. Median number of UMI counts ranged between 3990 and 10,176 per cell. Cells with extremely low number of UMI counts were filtered out using the default settings in Cell Ranger. Single-cell analysis was performed using Seurat (v.2.3.4) (Butler et al.[52]) applying default settings unless otherwise stated. Cells with >5% mitochondrial gene content and >5000 genes per cell were removed. Variables selected to be regressed out were nUMI and percent mitochondrial content. The irlba implementation PCA[53], was run and the first 20 principal components were used in subsequent analysis. Cells were clustered using Seurat's FindClusters function. To visualize the data, t-distributed stochastic neighbor embedding (t-SNE) plots were generated using the Seurat's RunTSNE function. All t-SNE plots displayed are on data not regressed for cell cycle, with the exception of Fig. 5d, e. Contaminants were identified based on signature gene expression and clusters were removed from the analysis (clusters 10, 11, and 12 from Supplementary Fig. 2c and clusters 11 and 14 from Supplementary Fig. 6d). Such cells formed their own clusters and appeared distinct from the main progenitor populations on t-SNE. Contaminant cells were electronically removed prior to subsequent analysis. Seurat's FindClusters function was used to cluster the remaining data using default perplexity. Differentially expressed genes between clusters were identified using Seurat's FindMarkers function with default settings: only genes showing expression in at least 10% of cells in one of the groups compared were considered; p-values were determined using the Wilcoxon ranked sum test; and genes having a Bonferonni-adjusted p-value of >0.05 were discarded. A generalized linear model[54] was used to determine significance. Violin plots shown in figures were made using Seurat's normalized data. Additional analysis and visualization were done using R[51]. The adult WT and FoxN1MycTg scRNA data set was visualized using the same analytical tools.

**Generation of gene scores**. We used the Seurat CellCycleScoring function to create scores for Ribosome using genes from the Kegg Ribosome gene set, or scores for Proliferation using a combined list of G2M and S-phase genes using a previously published marker set[20]. Fetal-specific genes were extracted using FindMarkers function comparing E13.5 cells against NB cells that had low proliferation scores. This initial list was filtered to remove genes for ribosomal proteins and remaining cell cycle genes, and further filtered to remove genes upregulated more than twofold in adult FoxN1MycTg cTEC population compared with adult WT cTEC in the bulk RNA-seq data. The fetal gene list was then used to similarly create a score using the Seurat CellCycleScoring function. Myc pathways were taken from MSigDB[49,55], namely DANG_MYC_TARGETS_UP and DANG_MYC_TARGETS_DOWN[56], Myc Max Targets[57].

**Intrathymic transplantation**. TEC enriched thymus samples from adult CD45.2 + WT and FoxN1MycTg mice were prepared as described above. The frequency of CD45− EpCAM + TEC sample was determined by flow cytometry prior to injection. WT TEC numbers injected were 166,000 (experiment 1) and 200,000 (experiment 2) per mouse. FoxN1MycTg TEC numbers injected were 133,000 (experiment 1) and 200,000 (experiment 2) per mouse. Both experiments gave similar results and so data were pooled. Cells were kept at 4 °C prior to injection. Intrathymic injections were performed as previously described[58] into CD45.1 + WT hosts, 6–10 weeks in age, that had received 500 Rad irradiation 2 h prior to surgery. Mice were anesthetized by intraperitoneal injection of Ketamine (100 mg/ml stock) (VetOne) plus Xylazine (20 mg/ml stock) (Akorn Animal Health) and provided with Buprenorphine (PAR Pharmaceutical) 0.5–2.0 mg/kg subcutaneous pre and postoperatively. Ten microliters of cells in suspension were injected into one thymic lobe using a Hamilton syringe equipped with a 30-gauge needle (Fisher Scientific). One or three wound clips (Fisher Scientific) were applied to close the incision. Mice were sacrificed 7–10 weeks post injection and successful injections were determined by the presence of CD45.2+ donor cells in the host CD45.1+ thymus. Successful engraftment was measured as an increase in total thymic cellularity and TEC number. Total counts and TEC samples were then stained for flow cytometric analysis as described above.

**Tamoxifen treatment**. Tamoxifen (Letco Medical) was dissolved in corn oil (Sigma C8267-500ml) at a concentration of 20 mg/ml by shaking 6 h at 37 °C. Adult mice were administered with 100 μl tamoxifen (2 mg per mouse) via intraperitoneal injection daily, for 5 consecutive days. Mice were sacrificed 25 weeks⁺ post treatment.

**Statistics**. Statistical significance was performed with GraphPad Prism. Differences between groups of mice were determined by a two-tailed unpaired Student's *t* test. If the F test to compare variance had a *p* value < 0.05 an unpaired *t* test with Welch's correction was performed, where we do not assume equal standard deviation.

**Reporting summary**. Further information on research design is available in the Nature Research Reporting Summary linked to this article.

## Data availability

All bulk RNA-seq data and scRNA-seq data is accessible GSE131368. The source data underlying Figs. 2a, c, 3f, 4a, c, d, 6a, b, c, 7c and f and Supplementary Figs. 4c, d, e, 5a, b, c, d, e and f are provided as a Source Data file.

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

## Acknowledgements

We thank Drs. David Levens, Dan Larson, Remy Bosselut, Sam John, Stefan Muljo, Joana Vidagali, Paul Love, Nancy Manley, and Ranjan Sen for critical comments and valuable discussions, Georg Holländer for sharing the *FoxN1Cre* mice, Jyoti Sen for providing aged mice, Ross Lake for help with imaging, Lauren R. Brinster for pathology assessment and the Center for Cancer Research Sequencing Facility and the CCR Flow Cytometry Core Facility for technical support. This work utilized the computational resources of the NIH HPC Biowulf cluster. (http://hpc.nih.gov) and R workbooks developed on the Palantir Foundry Platform (https://www.palantir.com/). This research was supported by the Intramural Research Program of the Center for Cancer Research at the National Cancer Institute, and by the NCI-NIA Joint Fellowship on Cancer and Aging.

## Author contributions

J.E.C. designed the research and performed the experiments, alongside Y.Z. J.E.C., J.M., M.C., and A.B. analyzed the data and produced the figures. C.H., M.O.S, M.K., and J.W. Y. provided reagents, mouse models and technical support. I.O. and Y.T. provided bulk RNA data. J.E.C. and A.B. directed and oversaw the experiments. All authors helped to design the research and read and commented on the paper.

## Competing interests

The authors declare no competing interests.
