## [Peer Review File · Nature Communications]

Reviewers' comments:

Reviewer #1 (Thymic development)(Remarks to the Author):

In this manuscript, Cowan et al examine heterogeneity within the thymic epithelial cell (TEC) compartment of mice, and examine mechanisms that regulate thymus growth across the life course. TEC are known critical regulators of T-cell development, and regulated patterns of thymus growth are important to sustain T-cell development, However, functional heterogeneity within TEC, and the mechanisms that control thymus growth, are poorly understood. Consequently, the scope of the study is relevant and of high interest.

The authors use bulk and scRNAseq to analyse the TEC makeup in both fetal and adult stages. This provides the first comprehensive analysis of TEC heterogeneity across the life course. They also identify Myc as a key regulator of fetal thymus, and in a series of elegant experiments gain and loss of function experiments, demonstrate its importance in the regulation of thymus growth and function. Collectively, the study is well performed, and provides insight into several aspects of thymus function that advances the field. I have some suggestions that would be of interest to pursue that are based on the data provided.

1. In Myctg mice, the authors clearly show that expression of Myc in TEC results in a larger thymus that contains more thymocytes. This includes mature CD4+ and CD8+ thymocytes. Are these increases caused by increased intrathymic proliferation, or production? Can the authors perhaps include CD62L/CD69, and proliferation analysis to further break down the SP thymocyte populations into immature and mature cells to address this?
2. Are Foxp3+ T-reg numbers also increased in the thymus and spleen of Myctg mice?
3. In Myc tg mice, both cTEC and mTEC are increased in numbers, which is due at least in part to increased proliferation in these cells. As mTEC are heterogeneous and can be divided into mTEC_{lo} and mTEC_{hi} subsets using MHCII and CD80, can the authors look to see if both mTEC subsets are equally increased? Along the same lines, as Aire+ mTEC are thought to be post-mitotic, can the authors analyse Aire expression in either thymic sections or by flow cytometry to see whether the Aire+ subset of mTEC is also increased in number?

Reviewer #2 (Systems immunology, thymocytes)(Remarks to the Author):

The manuscript by Cowan et al. identifies Myc as an important regulator of foetal thymic epithelial cell (TEC) proliferation and demonstrates that its forced constitutive or induced expression in post-natal TEC results in the adoption of several transcriptional and functional features typical of early life TEC. These changes correlate with an increase TEC cellularity and the cells' capacity to be transplantable when taken from adult donors and orthotopically grafted into adult mice.

The manuscript presents an interesting dataset and makes some very novel suggestions as to the importance of Myc in TEC biology. However, there are some significant limitations of study's design and analysis:

1. The analysis of "cTEC" and "mTEC" is likely to capture both true differential gene expression driven by cellular differences and differences in cTEC/mTEC subset proportions. This is particularly true for the bulk RNA-seq analysis. However, in the single cell RNA-seq analysis, there is no clear attempt to separate analysis by cTEC or mTEC subset as identified by cluster analysis. Similarly, the authors

should attempt to analyse the relative size of different cTEC/mTEC subsets between WT and Tg animals. The methods section, which is not easy to follow, describes an atypically method for analysing bulk RNA-seq data and would clearly benefit from a justification why this approach was chosen. There are packages that enable differential expression analysis of bulk RNA-seq data sets (DESeq2, EdgeR, Limma/Voom etc.), allow to fit a linear model to the data and can thus identify those gene that are differentially expressed as a function of age. The authors should provide further explanation why they chose to use filtering and why this was done for only those genes that monotonically increase or decrease.

2. In figure 1A (PCA plot), all points should be shown for each biological group and the description in the text should be reworded for clarity. All samples should be shown on the plot .

3. For bulk RNA-seq analysis, the authors use a cut-off of 1.5-fold. What is the basis for selecting this threshold?

4. Did the authors use ERCC spike-ins? Given the differences in RNA abundance, it may be important to explore this as a method of controlling for RNA abundance in the bulk RNA-seq analysis.

5. Thresholding single cells by high and low aggregate scores is reasonable but the authors should consider using latent variable analysis or similar general linear modelling with ribosomal or cell cycle as factors.

6. A few times throughout the manuscript, the authors use quite descriptive phrases in the results (e.g. "because this decline is evident in almost every cell at each developmental time point") without detailing the evidence that underlies this. They should include statistics/evidence to back up these sorts of statements. For example, the authors state that the FoxN1MycTg thymi have "comparably sized medullary areas observed by H&E staining (supplementary Figure 4A)." This should be quantified or they should make a more general statement stating that there were no obvious defects in thymic architecture.

7. The manuscript concludes that Myc protein availability is post-transcriptionally regulated. Are the authors aware of any miRNA that may act in such a way and – if so – could this be tested? This issue is not sufficiently explored but relevant as the expression of Myc is tightly regulated and RNA and protein have extremely short half-lives. For example (PMID: 19029303), the RNA-binding protein Igf2bp1 stabilizes Myc RNA via the Coding Region instability Determinant (CRD). Its post-natal decrease or loss of expression would impact on Myc protein availability and possibly half-life. Perhaps the authors could investigate the upstream transcriptional regulators of the differentially expressed genes and comment on factors driving the downregulation of Myc as a function of age. Myc is in the middle of a signalling cascade acting as a master regulator of growth but also influencing apoptosis, yet it is regulated by other pathways such as Notch.

8. An extended discussion of factors that regulate Myc stability would sharpen the understanding of the role that Myc plays both in TEC development and ageing.

9. Given their described mismatch between Myc RNA and MYC protein, it is slightly difficult to interpret an overexpression Myc model. The authors should comment on this point?

10. The FACS data in figure 4D demonstrates two cTEC populations that differ significantly in their UEA1 reactivity. The authors should comment on this finding and verify that this is indeed a regular staining pattern?

11. Regarding the data of peripheral T cells in constitutive and inducible Myc Tg mice, the manuscript would be hugely strengthened by a more detailed analysis of this compartment such as a quantification of naïve/memory/RTEs.

12. The authors state that "The FoxN1MycTg cTEC and mTEC samples maintained their lineage specific transcriptional identity". It is unclear to this reviewer what precisely is meant, how this identity was assessed and whether this was due to an inference based on data that appears to have a significant batch issue.

13. The tSNE plot in figure 5A appears to be dominated by batch effect. Has this been accounted for by canonical correlation analysis or nearest neighbour algorithms? This would be necessary before much biological inference can be placed in the cluster analysis. Indeed, a description and discussion of

batch correction or any attempt to assess the dominance of batch in the analysis/methods is missing. The clustering analysis seems to be largely extraneous and used only to remove non-TEC captured erroneously. Therefore, the clustering analysis would either need to be removed altogether or more extensively explored.

14. The authors note that "Interestingly, many of the pathways upregulated and downregulated in the FoxN1MycTg cTEC and mTEC samples displayed correspond with the pathways enriched and declining in WT TEC with age shown in Figure 1B and C." The authors should elaborate on this point which incompletely detailed in the present manuscript.

15. The manuscripts states "Interestingly, some fetal genes were not rescued by transgenic Myc expression (Supplementary Figure 5B), suggesting transgenic Myc only conferred expression of a limited set of fetal enriched genes and offering an explanation as to why adult FoxN1MycTg did not cluster with the E17.5 cTEC cells (Figure 5A)" This could be directly tested based on the PCA matrix acting as input to tSNE. However, this reviewer is not sure what this analysis adds furthermore to the differential analysis (which should capture almost the same genes).

16. The authors have not, as far as this reviewer can see, looked at tissue restricted antigen expression in any TEC subset. Could this information be include or, alternatively, explained why this is not possible? In this context, the authors may wish to comment whether Myc TG mice display features of autoimmunity.

17. Figure 5c shows reduced CD74 and H2-Aa expression in transgenic mTEC, suggesting a difference in sub-population structure in mutantant animals. Following batch correction, the sub-population structure of the WT and Tg mTEC should be investigated.

18. How do the authors explain the differences between the scRNA-seq cell cycle inference and BrdU results? How does this data compare when using other existing methods of cell cycle state inference such as Seurat's CellCycleScoring function or Cyclone.

19. Individual experiments should be shown for the TEC engraftment experiments. The methods section mentions that different (and differential) numbers of TEC were transferred and so readers should be able to assess how similar the results of these experiments were. Moreover, could the authors comment on the ability of adult CyclinD1 Tg TEC to engraft after transfer - should this analysis have been made – and qualify differential gene expression in TEC of these animals when compared to Myc Tg TEC as this will further identify pathways that may be of interest relating to thymic regeneration.

20. Error bars should be shown on bar-plots. I'm not sure why some are missing, but it appears that Figures 1c, 3e, 6b, S3 are lacking error bars.

The following minor issues should be addressed:

1. Typo: "acFScounting"
2. Typo: "Contaminates were removed from analysis[...]"
3. Figure 4 legend type: "C" should be "D"

In our detailed response to the Reviewers below, we have separated the Comments of each Reviewer in italics from our Response, and then provided our Revisions. We wish to thank the Reviewers for their detailed and insightful comments. We believe that accommodating their requests greatly improved our work.

Reviewer #1

In this manuscript, Cowan et al examine heterogeneity within the thymic epithelial cell (TEC) compartment of mice and examine mechanisms that regulate thymus growth across the life course. TEC are known critical regulators of T-cell development, and regulated patterns of thymus growth are important to sustain T-cell development. However, functional heterogeneity within TEC, and the mechanisms that control thymus growth, are poorly understood. Consequently, the scope of the study is relevant and of high interest.

The authors use bulk and scRNAseq to analyse the TEC makeup in both fetal and adult stages. This provides the first comprehensive analysis of TEC heterogeneity across the life course. They also identify Myc as a key regulator of fetal thymus, and in a series of elegant experiments gain and loss of function experiments, demonstrate its importance in the regulation of thymus growth and function. Collectively, the study is well performed, and provides insight into several aspects of thymus function that advances the field. I have some suggestions that would be of interest to pursue that are based on the data provided.

1. In Myctg mice, the authors clearly show that expression of Myc in TEC results in a larger thymus that contains more thymocytes. This includes mature CD4+ and CD8+ thymocytes. Are these increases caused by increased intrathymic proliferation, or production? Can the authors perhaps include CD62L/CD69, and proliferation analysis to further break down the SP thymocyte populations into immature and mature cells to address this?

Response:

We're grateful for the Reviewer's positive assessment of our work, and thank the Reviewer for this suggestion.

Revision:

To examine if the increase in observed CD4+ and CD8+ single-positive (SP) thymocytes was a consequence of increased intrathymic proliferation at the single positive stage, we performed a single BrdU 18hr pulse and measured the frequency of BrdU+ immature (CD69+ CD62L low) and mature (CD69- CD62L+) single positive thymocytes. (**Fig. S4E and Fig. S4F**). We found that the frequency of BrdU+ proliferating thymocytes in the mature CD4+ SP and CD8+ SP subsets was not increased in FoxN1MycTg thymi. Hence the increase in numbers of CD4+ SP and CD8+ SP thymocytes is not due to increased proliferation at the SP stage but reflects increased production.

Interestingly, we also found that the FoxN1MycTg mice have increased frequencies of mature CD69- CD62L+ thymocytes in both the SP4 and SP8 populations, compared to the WT littermate controls, suggestive of reduced thymic emigration. However, in spite of this possible decrease in emigration, we find that the enlarged thymi of FoxN1MycTg mice have increased function, with increased frequencies of recent thymic emigrants in the spleen (**Fig. S5F**).

2. Are Foxp3+ T-reg numbers also increased in the thymus and spleen of Myctg mice?

Response:

Because intra-thymic T-reg development is dependent upon mTEC interactions, we agree that alterations in development of this non-conventional lineage in this large thymus model are important to assess, as they may be indicative of disrupted mTEC development. Moreover, alterations to numbers of peripheral T-reg cells offers insight into possible peripheral consequences of increasing the thymus size to this scale.

Revision:

To examine if the FoxN1MycTg mice have altered numbers of T-reg we stained SP4 thymocytes and peripheral CD4 T cells for FoxP3 and CD25 (**Fig. S4D and Fig. S5E**). The results indicated a small but consistent increase in the frequency of CD25+ FoxP3+ Treg in the FoxN1MycTg thymus compared to WT (**Fig. S4D**). In contrast, the frequency of Treg was reduced in the spleen of the FoxN1MycTg; however, the overall increase in the numbers of total CD4 T cells within the spleen (**Fig. S5B**) resulted in no significant increase in numbers of T-reg in the spleen of the FoxN1MycTg mice (**Fig. S5E**).

3. In Myc tg mice, both cTEC and mTEC are increased in numbers, which is due at least in part to increased proliferation in these cells. As mTEC are heterogeneous and can be divided into mTEC^{lo} and mTEC^{hi} subsets using MHCII and CD80, can the authors look to see if both mTEC subsets are equally increased? Along the same lines, as Aire+ mTEC are thought to be post-mitotic, can the authors analyse Aire expression in either thymic sections or by flow cytometry to see whether the Aire+ subset of mTEC is also increased in number?

Response:

We thank the Reviewer for this suggestion. The transgenic expression of Myc in TEC resulted in a dramatic increase in the number of cTEC and mTEC in these large thymi. We agree that a more detailed analysis of how this effects mTEC heterogeneity is needed.

Revision:

To examine if the FoxN1MycTg thymus has an equal increase in mTEC^{hi} and mTEC^{low} subsets we looked at the frequency of each population by flow cytometry, as directed by the Reviewer. We detected no significant differences between the frequency of CD80+ Class II^{hi} mTEC and CD80- Class II^{low} mTEC (**Fig. 4D**). As the overall number of mTEC was significantly increased in FoxN1

MycTg, the absolute number of both the mTEChi and mTEClow subsets was significantly increased compared to WT littermates. Furthermore, intracellular staining of TEC for Aire expression revealed no changes to the frequency of Aire⁺ and Aire⁻ fractions of TEC within the CD80⁺ ClassII^{hi} mTEC subset (**Fig. 4D**). Again, due to the large increase in overall mTEC numbers, the number of both Aire⁺ and Aire⁻ mTEChi subsets was significantly increased compared to controls.

Reviewer #2

The manuscript by Cowan et al. identifies Myc as an important regulator of foetal thymic epithelial cell (TEC) proliferation and demonstrates that its forced constitutive or induced expression in post-natal TEC results in the adoption of several transcriptional and functional features typical of early life TEC. These changes correlate with an increase TEC cellularity and the cells' capacity to be transplantable when taken from adult donors and orthotopically grafted into adult mice.

The manuscript presents an interesting dataset and makes some very novel suggestions as to the importance of Myc in TEC biology. However, there are some significant limitations of study's design and analysis:

1a. The analysis of "cTEC" and "mTEC" is likely to capture both true differential gene expression driven by cellular differences and differences in cTEC/mTEC subset proportions. This is particularly true for the bulk RNA-seq analysis. However, in the single cell RNA-seq analysis, there is no clear attempt to separate analysis by cTEC or mTEC subset as identified by cluster analysis. Similarly, the authors should attempt to analyse the relative size of different cTEC/mTEC subsets between WT and Tg animals.

Response:

We're grateful for the Reviewer's enthusiasm for our work, and we appreciate the Reviewer's suggestion to compare subsets of cTEC/mTEC, and to analyse the relative size of different cTEC/mTEC subsets between WT and FoxN1MycTg animals. We agree that this will better reveal transcriptional changes in the scRNA data of the FoxN1 MycTg cTEC/mTEC samples compared to controls.

Revision:

Although subsets of fetal and adult cTEC are not established, we included cluster analysis to identify potential subsets of fetal and adult TEC using scRNAseq data (**Fig. S2B**). The cluster analysis revealed heterogeneity within the cTEC populations at E15.5 and NB based on dramatic differences in the proliferation score between clusters originating from the same developmental time point (**Fig. S2E**). In contrast, the ribosomal score was uniform in all clusters from the same aged samples. (**Fig. S2F**). Moreover, we performed cluster analysis of the scRNA seq data of WT and FoxN1MycTg samples, and focused on the recently identified mTEC I, II III and IV (pre-Aire, Aire, post-Aire, and Tuft cell) subsets (**Fig. 5 and Fig. S8**)(Bornstein et al., 2018). The increases in expression of ribosomal genes that we reported downstream of constitutive Myc expression remained evident in all subsets (**Fig. S8C**), whereas increases in expression of cell cycle genes were evident in 3 of these 4 subsets (**Fig. S8B**). We're grateful to the Reviewer for suggesting these analyses, as their inclusion further strengthens our study.

1b. The methods section, which is not easy to follow, describes an atypically method for analysing bulk RNA-seq data and would clearly benefit from a

justification why this approach was chosen. There are packages that enable differential expression analysis of bulk RNA-seq data sets (DEseq2, EdgeR, Limma/Voom etc.), allow to fit a linear model to the data and can thus identify those gene that are differentially expressed as a function of age. The authors should provide further explanation why they chose to use filtering and why this was done for only those genes that monotonically increase or decrease.

Response:

We apologize for the lack of clarity in the Materials and Methods section of our previous submission. We indeed performed the standard analytical Limma pipeline described here by the author for nearly all analyses, but we did not describe it well.

Revision:

We have amended the text to offer more clarity on the techniques used to analysis both the scRNA and the bulk RNA seq data set. The atypical analysis previously used for Supplementary Figure 1C has been removed. We clarify that the 2000 most differentially expressed genes across all timepoints were based on p-values using the limma package from the bulk RNA-seq data prior to running GSVA to find the most enriched pathways (See Materials and Methods).

2. In figure 1A (PCA plot), all points should be shown for each biological group and the description in the text should be reworded for clarity. All samples should be shown on the plot.

Response and Revisions

We agree. We have amended the PCA in Figure 1A to show all replicas per biological group. Furthermore, we added two additional PCA plots for cTEC or mTEC samples only (**Fig. S1D**). We think this better explains how the pathway enrichment analysis was performed in Figure 1B and 1C.

3. For bulk RNA-seq analysis, the authors use a cut-off of 1.5-fold. What is the basis for selecting this threshold?

Response and Revision:

We agree that this earlier analysis was not optimal. This analysis is now represented as Venn diagrams, displaying the number of lineage specific genes and overlapping genes significantly increasing or decreasing in expression with age in mTEC and cTEC lineages (**Fig. S1C**). A fold-change cutoff of 2 together with a p-value cutoff of 0.05 is now used, similar to what is commonly used in many publications. The same analysis was performed on the significantly altered genes changing in expression between the adult and aged samples (**Fig. S1E**), and upregulated and downregulated in the FoxN1 MycTg cTEc and mTEC samples compared to WT (**Fig. S6A**).

4. Did the authors use ERCC spike-ins? Given the differences in RNA abundance, it may be important to explore this as a method of controlling for RNA abundance in the bulk RNA-seq analysis.

Response: We consulted with Dr. Marc Salit (at Stanford) who is an authority on ERCC spike-ins and learned that there is not agreement on the use of ERCC

spike-ins when very small numbers of cells are assessed by RNA-seq, as was necessary in our experiments. In addition, when planning this study in 2015, we performed preliminary experiments to assess the feasibility of ERCC spike-ins. In these experiments, normalization by ERCC spike-in led to >2-fold differences in measured mRNA abundance between replicate RNA-seq experiments that were analyzed alongside each other, and in some cases the differences between replicates was even larger. However, when analyzed conventionally, without normalization to spike-in controls, the replicate samples gave very similar results to each other. This may be because when we isolate very small numbers of cells, in the order of a few thousand, it can be difficult to accurately determine the exact number of cells. Additionally, efficiencies of RNA extraction from cells will differ between samples. This is cogently discussed on the Bioconductor support website (<https://support.bioconductor.org/p/88413/>) in a post by Dr. Simon Anders, University of Heidelberg.

In addition, in some samples of our preliminary experiments, far too many of the sequences obtained were of the ERCC spike-ins, which meant that many fewer sequences were obtained from the sample of interest. Hence the ERCC spike-ins would have greatly increased both the cost as well as the noise in our experiments.

One further point is that we show that total RNA also declines during development (**Fig. 2C**). Hence, the decline in Myc activity with developmental age that we report is likely to be an underestimate if absolute amounts of mRNA are considered i.e. our conclusions are not altered by the absence of ERCC spike-in controls. We therefore do think that the major conclusion presented in our manuscript is robust.

Revision: We now describe our reason not to use ERCC spike-ins because of the lack of a standard methodology for small numbers of cells (Materials and Methods). We mention that the decline in Myc activity may be even larger than we report, if absolute RNA amounts per cell are considered.

5. Thresholding single cells by high and low aggregate scores is reasonable but the authors should consider using latent variable analysis or similar general linear modelling with ribosomal or cell cycle as factors.

Response:

We thank the Reviewer for their suggestion and agree. We linearly regressed ribosomal gene expression on cTEC developmental age and cell proliferation score. Here, we find that cell age is a much more robust predictor of ribosomal gene expression than is cell proliferation score ($R^2 = 0.67$ vs $R^2 = 0.11$). This is reinforced by the partial regression analysis, which finds that the bulk of the explanatory power of cTEC age is independent of cell proliferation score (proliferation score-independent $R^2 = 0.62$). We modeled the impacts of cTEC age and cell proliferation score on ribosomal gene expression using a general linear model. Specifically, we used R's least squares regression function *lm*. Additionally, we calculated partial regression coefficients of determination for the two explanatory variables, that is, the fraction of the residual variance when regressing ribosomal score on cell cycle (alternatively, age) explained by age

(alternatively, cell cycle) using the *rsq* package in R. Results from multiple least squares regression analysis (**pls** package).

Model: ribosome score ~ age $R^2 = 0.67$

Model: ribosome score ~ cell cycle score $R^2 = 0.11$

Model: ribosome score ~ age + cell cycle score $R^2 = 0.67$

Results from partial least squares regression analysis (**rsq** package):

Age partial $R^2 = 0.62$

Cell cycle score partial $R^2 = 0.0004$

Revision:

We have added these results to the text describing Figure 3. We thank the Reviewer for the suggestion to include this analysis, as it strengthens our conclusion.

6. A few times throughout the manuscript, the authors use quite descriptive phrases in the results (e.g. “because this decline is evident in almost every cell at each developmental time point”) without detailing the evidence that underlies this. They should include statistics/evidence to back up these sorts of statements. For example, the authors state that the FoxN1MycTg thymi have “comparably sized medullary areas observed by H&E staining (supplementary Figure 4A).” This should be quantified or they should make a more general statement stating that there were no obvious defects in thymic architecture.

Response and Revision:

We apologize for the inappropriate use of descriptive phrases. We have changed the wording accordingly in these and similar sections.

7. The manuscript concludes that Myc protein availability is post-transcriptionally regulated. Are the authors aware of any miRNA that may act in such a way and – if so – could this be tested? This issue is not sufficiently explored but relevant as the expression of Myc is tightly regulated and RNA and protein have extremely short half-lives. For example (PMID: 19029303), the RNA-binding protein Igf2bp1 stabilizes Myc RNA via the Coding Region instability Determinant (CRD). Its post-natal decrease or loss of expression would impact on Myc protein availability and possibly half-life. Perhaps the authors could investigate the upstream transcriptional regulators of the differentially expressed genes and comment on factors driving the downregulation of Myc as a function of age. Myc is in the middle of a signaling cascade acting as a master regulator of growth but also influencing apoptosis, yet it is regulated by other pathways such as Notch.

8. An extended discussion of factors that regulate Myc stability would sharpen the understanding of the role that Myc plays both in TEC development and ageing.

9. Given their described mismatch between Myc RNA and MYC protein, it is slightly difficult to interpret an overexpression Myc model. The authors should comment on this point?

Response:

We have chosen to address point 7, 8 and 9 collectively, as we think the three issues are related.

We thank the Reviewer for these suggestions and insights. We agree that investigating the upstream transcriptional regulators of Myc is of considerable interest. This is something we plan to explore in the future. However, these are detailed and involved studies, as the Reviewer likely appreciates, and we could not complete such studies in the time we were given for these revisions. Indeed, our initial assessments suggest that multiple upstream controllers of Myc will need to be assessed, including Igf2bp1 mentioned by the Reviewer but also Igf2bp3, as well as Igf2 and perhaps other factors also, such as transcription factor Hmga2 (**Fig. S7A**). Such studies are likely to take several years.

The fact that overexpression of Myc was clearly evident in our studies using Foxn1MycTg mice is an important insight pointed out by the Reviewer in point 9, and very relevant for understanding factors that might be regulating Myc in TEC. The FoxN1MycTg mouse has a human MYC construct inserted into the Rosa 26 locus behind LoxP sites (Calado et al., 2012) that lacks the 5' UTR, 3' UTR or introns that are present in the endogenous mouse locus, that may be the targets of post-transcriptional control. Additionally, human and mouse Myc are ~89% conserved at the amino acid level, and amino acids that differ may be the site of post-translational modifications that affect protein stability. Either of these possibilities could result in the observed increase of Myc protein in the FoxN1MycTg TEC. Investigating possible alterations to post-transcriptional or post-translational modifications of this human MYC construct would be of considerable interest, as it would offer greater insight into how endogenous Myc is regulated in TEC through development. However, again these are detailed and involved studies and we could not complete such studies in the time assigned for these revisions. We agree that they are critical studies for future work, and we thank the Reviewer for this excellent insight.

Revision:

We have extended the discussion section to include possible factors that regulate Myc post-transcriptionally. Moreover, we have discussed the overexpression Myc model in greater detail in the Discussion. In addition, we have added the transcript levels of human MYC of the MycTg cTEC and mTEC bulk RNA seq samples (**Fig. S3A**).

10. The FACS data in figure 4D demonstrates two cTEC populations that differ significantly in their UEA1 reactivity. The authors should comment on this finding and verify that this is indeed a regular staining pattern?

Response and Revisions:

The data in Figure 4D were not typical of all of our experiments, and we have replaced them with more representative data.

11. *Regarding the data of peripheral T cells in constitutive and inducible Myc Tg mice, the manuscript would be hugely strengthened by a more detailed analysis of this compartment such as a quantification of naïve/memory/RTEs.*

Response:

We agree with the Reviewer that quantification of naïve/memory/RTEs T cells in the FoxN1MycTg mice would strengthen the manuscript, as it would establish that the large thymi of these mice have increased function. We had already initiated a cross of the constitutive MycTg strain to Rag2-GFP mouse, to enable detailed analysis of RTEs in the periphery. We obtained sufficient numbers of these mice aged 5 weeks that we could perform the experiments requested by the Reviewer within the time frame of these Revisions. We were not able to perform similar experiments with inducible MycTg mice in the necessary time frame, and indeed it will take many more months before we have Rag2-GFP mice crossed to inducible MycTg mice and can perform these additional experiments. We agree that such experiments on the inducible MycTg are well worth performing, together with additional experiments examining whether thymic regrowth can reverse or ameliorate the immune defects described in aging mice. In our experiments with the inducible Myc model we waited many months to allow thymic regrowth. Hence these additional studies will likely require years to perform, and we respectfully suggest they are beyond the scope of the present study. We are glad to include our data obtained with the constitutive MycTg mice, and we agree they greatly strengthen this study.

Revision:

We measured the frequency of GFP⁺ CD4 and CD8 T cells in the spleen of FoxN1MycTgRag2-GFP mice and saw an increase in RTEs (**Fig. S5F**). In addition to the assessment of RTE, we explored if the large thymi of FoxN1MycTg mice also resulted in altered proportions of naïve to memory T cell populations, as suggested by the Reviewer. Flow cytometric analysis revealed an increase in the numbers of CD62L⁺ CD44⁻ naïve CD4 and CD8 T cells in the FoxN1MycTg spleen compared to controls (**Fig. S5C and S5D**). Moreover, the FoxN1MycTg mice showed a significant increase in the number of CD62L⁺ CD44⁺ central memory CD8 T cells. No alteration in the numbers of CD62L⁺ CD44⁺ central memory CD4 T cells or CD62L⁻ CD44⁺ effector memory CD4 and CD8 T cells were observed in the FoxN1MycTg mice.

12. *The authors state that “The FoxN1MycTg cTEC and mTEC samples maintained their lineage specific transcriptional identity”. It is unclear to this Reviewer what precisely is meant, how this identity was assessed and whether this was due to an inference based on data that appears to have a significant batch issue.*

Response:

We apologize for the lack of clarity in our comment. As we do not think we have batch issues on the data set in Figure 5 (see response to point 13 below) we do not think this would interfere with the clustering of cTEC and mTEC populations from WT and FoxN1MycTg mice. We agree that our comment “The FoxN1MycTg

cTEC and mTEC samples maintained their lineage specific transcriptional identity” was poorly stated. We meant to state that cTEC and mTEC populations from the FoxN1MycTg mice cluster close to their WT counterparts by tSNE analysis (**Fig. 5 and Fig. S6C**) and continue to express selected lineage specific genes displayed as feature plots (**Fig. S6D**).

Revision:

We now state that cTEC and mTEC populations from FoxN1MycTg mice cluster close to their WT counterparts in the Results.

13. The tSNE plot in figure 5A appears to be dominated by batch effect. Has this been accounted for by canonical correlation analysis or nearest neighbour algorithms? This would be necessary before much biological inference can be placed in the cluster analysis. Indeed, a description and discussion of batch correction or any attempt to assess the dominance of batch in the analysis/methods is missing. The clustering analysis seems to be largely extraneous and used only to remove non-TEC captured erroneously. Therefore, the clustering analysis would either need to be removed altogether or more extensively explored.

Response:

We do not believe technical batch effect plays a determinative role in Figure 5, as all cell populations shown in this Figure were sequenced in the same batch. We apologize for having omitted this essential information in our original submission.

To confirm the validity of these results, we examined T cells included as contaminants in multiple populations, as these are expected to be identical, unlike the thymic epithelial cells. As can be seen in Figure S6C and the Figure below, T-cells (identified based on Cd3e and high Tcf7 expression) contaminating each of WT cTEC, TG cTEC, WT mTEC, and TG mTEC populations cluster tightly together. In addition, we did not think it is advisable to use batch correction tools. CellRanger’s aggr and Seurat version 3’s alignment function advise that there be at least one cell population present in both, or all, datasets to be integrated (Stuart et al 2019; Hagverdi et al 2018). The five cell populations in Figure 5, however, are all biologically distinct. Dr. Maggie Cam, the Head of the Collaborative Bioinformatics Resource, and Dr. Mike Kelley, the Director of our Single-Cell facility, are co-authors on this manuscript, and agree with this assessment.

Revision:

We are of course willing to further revise this section if the Reviewer directs us to.

Figure 1 for Reviewers: scRNA-seq analysis was performed on cTEC (CD45⁺EpCAM⁺Ly51⁺UEA⁻) or mTEC (CD45⁺EpCAM⁺Ly51⁻UEA⁺) isolated from adult WT or FoxN1MycTg mice, and a cTEC sample from embryonic day 17.5 (E17.5 cTEC) from the experiment shown in Figure 5 of the manuscript. CD3⁺ cells that contaminate each of the 4 adult populations cluster tightly together in Tsne analysis.

14. The authors note that “Interestingly, many of the pathways upregulated and downregulated in the FoxN1MycTg cTEC and mTEC samples displayed correspond with the pathways enriched and declining in WT TEC with age shown in Figure 1B and C.” The authors should elaborate on this point which incompletely detailed in the present manuscript.

Response and Revision:

We apologize for our lack of clarity. The comment was intended to highlight the overlap between the pathways included in the heat map in Figure 1B and 1C to the pathways listing in Figure S6B. We have amended the Results to clarify which specific pathways we are referring to.

15. The manuscripts states “Interestingly, some fetal genes were not rescued by transgenic Myc expression (Supplementary Figure 5B), suggesting transgenic Myc only conferred expression of a limited set of fetal enriched genes and offering an explanation as to why adult FoxN1MycTg did not cluster with the E17.5 cTEC cells (Figure 5A)” This could be directly tested based on the PCA matrix acting as input to tSNE. However, this Reviewer is not sure what this analysis adds furthermore to the differential analysis (which should capture almost the same genes).

Response:

We agree with the Reviewer, and believe our statement was unclear. The important point we wanted to make was that there is a set of genes highly expressed near-exclusively in fetal cells, whose expression is not restored by

transgenic Myc i.e. Myc is one component of the fetal transcriptional program, but not all.

Revision:

We now simply state that “some fetal genes were not rescued by transgenic Myc expression (**Fig. S7A and Fig. S7B**), indicating that transgenic Myc represented only one component of the fetal transcriptional program as it only conferred expression of a limited set of fetal enriched genes.

16. The authors have not, as far as this Reviewer can see, looked at tissue restricted antigen expression in any TEC subset. Could this information be include or, alternatively, explained why this is not possible? In this context, the authors may wish to comment whether Myc TG mice display features of autoimmunity.

Response:

We thank the Reviewer for this important point. We agree that it is of high importance to confirm that transgenic expression of Myc in TEC does not alter the expression of tissue-restricted antigen expression in any mTEC subset, which may consequently result in the FoxN1MycTg mice presenting with signs of autoimmunity.

Revision:

To address the suggestion by the review we assessed the expression of Aire dependent genes in the scRNA seq data set (**Fig. 5G**). We clustered the populations to separate the WT and FoxN1MycTg mTEC populations into the four recently defined mTEC subsets (**Fig. 5D and Fig. S8A**) (see point 17 below) (Bornstein et al., 2018). We confirmed there were no alterations in the expression of Aire dependent or independent genes in any of the four mTEC populations between the WT and FoxN1 MycTg samples (**Fig. 5G**). This result indicates transgenic expression of Myc in TEC does not alter the expression of tissue-restricted antigen expression. Moreover, to confirm if the FoxN1MycTg mice display signs of autoimmunity, we harvested multiple organs (liver/lung/kidney/intestine/skin) from young adult FoxN1MycTg and performed H and E staining. We sent the sections to a pathologist (Dr. Lauren Brinster, V.M.D., NIH) who confirmed they appeared similar to the littermate WT controls (not shown). Additionally, we transferred CD4 and CD8 T cells from young FoxN1MycTg and WT controls into immunodeficient NOD scid gamma (NSG) hosts and confirmed no signs of autoimmunity in any of the hosts (data not shown). However, we think this interesting question should be examined further in future work in the context of aging FoxN1MycTg mice. Indeed, one hypothesis raised by the Reviewer’s question is that age-related thymic involution might offer an advantage to the aging host by reducing the risk of autoimmunity. Future studies along the lines suggested by the Reviewer may provide evidence for or against this hypothesis.

17. Figure 5c shows reduced CD74 and H2-Aa expression in transgenic mTEC, suggesting a difference in sub-population structure in mutant animals. Following

batch correction, the sub-population structure of the WT and Tg mTEC should be investigated.

Response:

As mentioned, we do not think the data in Figure 5 are affected by batch issues. We agree that an examination of the sub-population structure of the WT and Tg mTEC would improve the manuscript.

Revision:

To address this suggestion by the Reviewer, we focused our analysis on the mTEC populations from the scRNA-seq data set and clustered these samples at a resolution that enabled the identification of the four new defined mTEC subsets (**Fig. 5E and Fig. S8A**). This cluster analysis confirmed that each of the four clusters contained both WT and FoxN1MycTg cells (**Fig. 5D and Fig. 5E**). We calculated the frequency of total mTEC from WT or FoxN1MycTg samples that made up each subset which revealed the FoxN1MycTg mTEC had no major disruption to the frequencies of these four subpopulations compared to WT (**Fig. 5F**). An increase in the frequency of mTEC IV Tuft cells was observed in the FoxN1MycTg cells. The relatively undisrupted mTEC heterogeneity of the FoxN1MycTg cells by scRNA-seq is supported by the flow cytometric data of mTEChi and mTEClow populations and the frequency of Aire⁺ and Aire⁻ mTEChi populations being unaltered in FoxN1 MycTg mice (**Fig. 4D**).

18. How do the authors explain the differences between the scRNA-seq cell cycle inference and BrdU results? How does this data compare when using other existing methods of cell cycle state inference such as Seurat's CellCycleScoring function or Cyclone.

Response:

We indeed used the Seurat's CellCycleScoring function to calculate the cell cycle score displayed in Figure 3B and Figure 5B and apologize that this was unclear in the Material and Methods section. We have re-examined our analysis of these results and we do not think there is a discrepancy, as the FoxN1MycTg TEC showed increased BrdU labeling (**Fig. 6A**) and increased cell cycle score (**Fig. 5B**).

Revision:

We have amended the Material and Methods section to clearly state that the cell cycle score was generated using the Seurat's CellCycleScoring function, combining the G2M and S score together, to generate one proliferation score. We apologize for the lack of clarity in our earlier submission.

19. Individual experiments should be shown for the TEC engraftment experiments. The methods section mentions that different (and differential) numbers of TEC were transferred and so readers should be able to assess how similar the results of these experiments were. Moreover, could the authors comment on the ability of adult CyclinD1 Tg TEC to engraft after transfer - should this analysis have been made – and qualify differential gene expression in TEC of these animals when compared to Myc Tg TEC as this will further identify pathways that may be of interest relating to thymic regeneration.

Response:

We agree with the Reviewer that the readers should be able to assess how similar the results of each intrathymic injection experiments were. We realized that one of our two experiments compared intrathymic injections of very different numbers of TEC (3×10^5 wild-type TEC, but 6.45×10^5 MycTg TEC). We have replaced this experiment with a different experiment we recently obtained, where we transplanted equal numbers of wild-type and MycTg TEC (2×10^5). We retained the experiment we previously presented that compared similar input TEC numbers (1.66×10^5 wild-type TEC versus 1.33×10^5 MycTg TEC). We changed the bar graphs to show the two different experiments.

It would indeed be of great interest to assess the ability of adult CyclinD1 Tg TEC to engraft after transfer. The bulk RNA seq data on the Cyclin D1 mice (**Fig. S7C and Fig. S7D**) was performed by our collaborators Izumi Ohigashi and Yousuke Takahama at the University of Tokushima, Japan. We do not maintain our own Cyclin D1 mouse colony and, therefore, we were unable to perform intrathymic transfer with CyclinD1 Tg TEC. We agree with the Reviewer that this experiment is well worth doing, and we also think that either result would be of interest and value. We do intend to establish our own cyclin D1 colony, so we can eventually better compare them for this and other points with FoxN1MycTg mice.

Revision:

We have changed the bar graph in Figure 7C to show individual experiments for the TEC engraftment experiments, indicating that Foxn1MycTg adult TEC are transplantable.

20. Error bars should be shown on bar-plots. I'm not sure why some are missing, but it appears that Figures 1c, 3e, 6b, S3 are lacking error bars.

Response and Revisions:

The relevant error bars and experimental details have been added for all of these figures, with the exception of 2C and 6B, where the WT adult samples are set to 1 and so have no error bar. We did this to allow us to compile experiments ran on different flow cytometers, which give a large range of PY MFI.

Thank you for noticing our mistakes.

The following minor issues should be addressed:

- 1. Typo: "acFScounting"*
- 2. Typo: "Contaminates were removed from analysis[...]"*
- 3. Figure 4 legend type: "C" should be "D"*

Response and Revisions:

We have corrected all of these errors, and we are grateful to the Reviewer for these corrections.

References:

- Bornstein, C., S. Nevo, A. Giladi, N. Kadouri, M. Pouzolles, F. Gerbe, E. David, A. Machado, A. Chuprin, B. Toth, O. Goldberg, S. Itzkovitz, N. Taylor, P. Jay, V.S. Zimmermann, J. Abramson, and I. Amit. 2018. Single-cell mapping of the thymic stroma identifies IL-25-producing tuft epithelial cells. *Nature* 559:622-626.
- Calado, D.P., Y. Sasaki, S.A. Godinho, A. Pellerin, K. Kochert, B.P. Sleckman, I.M. de Alboran, M. Janz, S. Rodig, and K. Rajewsky. 2012. The cell-cycle regulator c-Myc is essential for the formation and maintenance of germinal centers. *Nat Immunol* 13:1092-1100.

REVIEWERS' COMMENTS:

Reviewer #1 (Remarks to the Author):

Cowan et al have provided a revised version of their manuscript that addresses all of the comments I initially raised. They have provided new data to address multiple points, including detailed analysis of thymocyte proliferation and heterogeneity, Foxp3+ T-Reg development and mTEC heterogeneity. These revisions strengthen what is an interesting manuscript that will provide important new information to the field.

Reviewer #2 (Remarks to the Author):

The authors have answered all of the reviewer's comments.